# Phospholipid translocation captured in a bifunctional membrane protein MprF

Danfeng Song [1,2,3], Haizhan Jiao[1,2,3] & Zhenfeng Liu [1,2✉]

As a large family of membrane proteins crucial for bacterial physiology and virulence, the Multiple Peptide Resistance Factors (MprFs) utilize two separate domains to synthesize and translocate aminoacyl phospholipids to the outer leaflets of bacterial membranes. The function of MprFs enables *Staphylococcus aureus* and other pathogenic bacteria to acquire resistance to daptomycin and cationic antimicrobial peptides. Here we present cryo-electron microscopy structures of MprF homodimer from *Rhizobium tropici* (*Rt*MprF) at two different states in complex with lysyl-phosphatidylglycerol (LysPG). *Rt*MprF contains a membrane-embedded lipid-flippase domain with two deep cavities opening toward the inner and outer leaflets of the membrane respectively. Intriguingly, a hook-shaped LysPG molecule is trapped inside the inner cavity with its head group bent toward the outer cavity which hosts a second phospholipid-binding site. Moreover, *Rt*MprF exhibits multiple conformational states with the synthase domain adopting distinct positions relative to the flippase domain. Our results provide a detailed framework for understanding the mechanisms of MprF-mediated modification and translocation of phospholipids.

[1] National Laboratory of Biomacromolecules, CAS Center for Excellence in Biomacromolecules, Institute of Biophysics, Chinese Academy of Sciences, Beijing, PR China. [2] College of Life Sciences, University of Chinese Academy of Sciences, Beijing, PR China. [3] These authors contributed equally: Danfeng Song, Haizhan Jiao. ✉email: liuzf@ibp.ac.cn

Aminoacyl phospholipids, such as lysyl-phosphatidylglycerol (LysPG), are an important class of lipids with crucial biological functions in antibiotic resistance, pathogenicity, stress response, and motility of bacterial cells[1]. They are widely distributed in microbes such as *Staphylococcus aureus*, *Mycobacterium tuberculosis*, and many other pathogenic bacteria[2–4]. An integral membrane protein known as Multiple Peptide Resistance Factor (MprF) catalyzes biosynthesis of aminoacyl phosphatidylglycerol (aaPG) by using phosphatidylglycerol and aminoacyl-tRNA as substrates[5–7]. MprF orthologs from different species can modify phosphatidylglycerol (PG) or cardiolipin with distinct aminoacyl groups including lysyl, alanyl, arginyl, and ornithyl groups[1,5,8]. When *mprF* was knocked out in *S. aureus*, the bacteria became highly susceptible to daptomycin and cationic antimicrobial peptides (CAMPs) including defensins from human neutrophil and bacteriocins[9,10]. Daptomycin is a cyclic lipopeptide antibiotic of last resort for treating infections caused by methicillin-resistant and vancomycin-resistant *S. aureus* (MRSA and VRSA) as well as other multidrug-resistant gram-positive bacteria[11,12]. Numerous mutations of *mprF* have been identified in the daptomycin-resistant (DAP-R) *S. aureus* and among them, several gain-of-function mutations were verified as the causes of DAP-R phenotype[13–18].

MprF is a bifunctional protein with two separate domains and the antimicrobial peptide resistance of *S. aureus* requires the presence of both domains[5,9]. While the cytoplasmic domain of MprF functions as an aaPG synthase[19], the membrane-spanning domain serves as a phospholipid flippase mediating translocation of aaPG from the inner leaflet to the outer leaflet of the membrane[9]. Occurrence of LysPG in the outer leaflet of bacterial membrane might help to repel CAMPs from reaching the membrane surface through electrostatic repulsion[20], modulate the peptide–membrane interactions[21] or inhibit the formation of membrane leaks induced by the CAMPs[22]. As a widespread virulence factor causing CAMP and antibiotic resistance in pathogenic bacteria, MprF is considered as a promising target for development of anti-infective strategies against drug-resistant bacteria[23,24]. Although crystal structures of the synthase domain of MprF have been solved recently and provided preliminary insights into its substrate-binding sites[19], little is known about the mechanisms of aaPG translocation, interaction of MprF with antibiotics or coupling between the two domains, mainly due to lack of the full-length MprF structure. Here we present the cryo-electron microscopy (cryo-EM) structures of MprF from *Rhizobium tropici* (*Rt*MprF) at two different states, unraveling notable features related to LysPG recognition and translocation as well as antibiotic resistance.

## Results

**Overall structure and oligomeric state of *Rt*MprF.** Recombinant *Rt*MprF protein with a hexahistidine tag fused to its carboxyl-terminal region was expressed in *E. coli* cells, and the protein was purified through immobilized metal affinity chromatography in solutions with either *n*-dodecyl-β-D-maltoside (β-DDM) or glycodiosgenin (GDN) (see "Methods" for more details). For single-particle cryo-EM analysis, the purified *Rt*MprF protein was further reconstituted with PG into lipid nanodiscs, a nanoscale complex system consisting of a small patch of lipid bilayer and the target protein surrounded by engineered membrane-scaffold protein[25] (Supplementary Figs. 1a and 2a). Two-dimensional (2D) and three-dimensional (3D) classes of the single-particle images indicate that *Rt*MprF exists mainly as homodimers in nanodiscs (Supplementary Figs. 1b, c and 2b–d). The cryo-EM maps for *Rt*MprF(DDM)-nanodiscs and *Rt*MprF(GDN)-nanodiscs

(full-length *Rt*MprF protein purified in β-DDM/GDN and reconstituted in nanodiscs) were refined to 3.7 and 2.96 Å resolution, respectively (Supplementary Figs. 1d–f and 2e–g). The cryo-EM map of *Rt*MprF(GDN)-nanodiscs exhibits well-defined features allowing construction of a structural model with ~94.4% amino acid residues of the full-length *Rt*MprF protein and identification of four lipid molecules per monomer (Supplementary Fig. 3 and Table 1). The structure of *Rt*MprF(DDM)-nanodiscs contains three lipid molecules and represents a state different from that of *Rt*MprF(GDN)-nanodiscs as discussed below. The crystal structure of the catalytic domain of *Rt*MprF in the C-terminal region has been solved at 2.0 Å resolution (Supplementary Fig. 4), and serves as the initial model for building the corresponding region in the cryo-EM structures of the full-length *Rt*MprF.

While the MprF protein from *S. aureus* (*Sa*MprF) may oligomerize into homodimers or homotetramers[26], *Rt*MprF in nanodiscs mainly exists as an arch-shaped homodimer with the C2 symmetry axis running through the dimerization interface (Fig. 1a–d). The function of *Rt*MprF is related to polymyxin B (a lipopeptide antibiotic) resistance, acid tolerance, nodulation competitiveness of *R. tropici* under low pH conditions[27,28]. To analyze the oligomeric state of *Rt*MprF on the membrane, the *E. coli* membrane with recombinant *Rt*MprF protein was prepared and incubated with a bifunctional amine-reactive crosslinking reagent (disuccinimidyl suberate, DSS). After crosslinking, the products were solubilized in β-DDM solution, separated through sodium dodecyl sulfate–polyacrylamide gel electrophoresis (SDS-PAGE) and detected through western blot by using the anti-His-Tag antibody. The crosslinking result demonstrates that the dimeric form of *Rt*MprF protein is present in the membrane (Fig. 1e). Dimerization of *Rt*MprF is mainly mediated by the transmembrane domain and a total surface area of 12080.9 Å² is buried in the *Rt*MprF dimer with 4263.3-Å² interface area between two adjacent monomers, indicating the dimeric state of *Rt*MprF is stable in lipid nanodiscs, according to the result of quantitative analysis through PISA (proteins, interfaces, structures, and assemblies) program[29]. Curiously, four lipid molecules, namely PG1, PG2, and their symmetry-related molecules PG1′ and PG2′, are located at the monomer–monomer interface (Fig. 1f). They collectively contribute to dimerization of *Rt*MprF by forming polar and hydrophobic interactions with two adjacent monomers simultaneously. While the protein–protein contact within the dimer contributes merely 629.5-Å² interface area, the four PG molecules form 1646.3 and 1643.1-Å² interface area with the two adjacent monomers, respectively, suggesting that these interfacial lipid molecules have crucial roles in stabilizing the dimeric state of *Rt*MprF. In *Rt*MprF(GDN)-nanodisc, the hydrophobic group of a GDN molecule occupies the binding site of the 2-acyl chain of PG2 molecule observed in *Rt*MprF(DDM)-nanodisc, while the other PG molecule (PG3) at the peripheral region of the dimerization interface is located nearby GDN (Supplementary Fig. 3c).

***Rt*MprF consists of a membrane-embedded flippase domain interacting closely with the synthase domain.** Each *Rt*MprF monomer contains a large membrane-spanning flippase domain with 12 long transmembrane helices (TM1–2, TM4–6, TM8–14) and two pairs of short α-helices (TM3a–TM3b and TM7a–TM7b) spanning halfway in the membrane (Fig. 2a–c). The cytoplasmic surface of the flippase domain mainly carries positive electrostatic potential, whereas the periplasmic surface is mostly covered by negative potential (Supplementary Fig. 5a). The flippase domain can be divided into two subdomains, namely Subdomains 1 and 2 (Fig. 2b). Subdomain 1 covers the region from Leu24 to Arg325 and contains TM1, 2, 3a, 3b, 4–6, 7a, 7b, and TM8, whereas

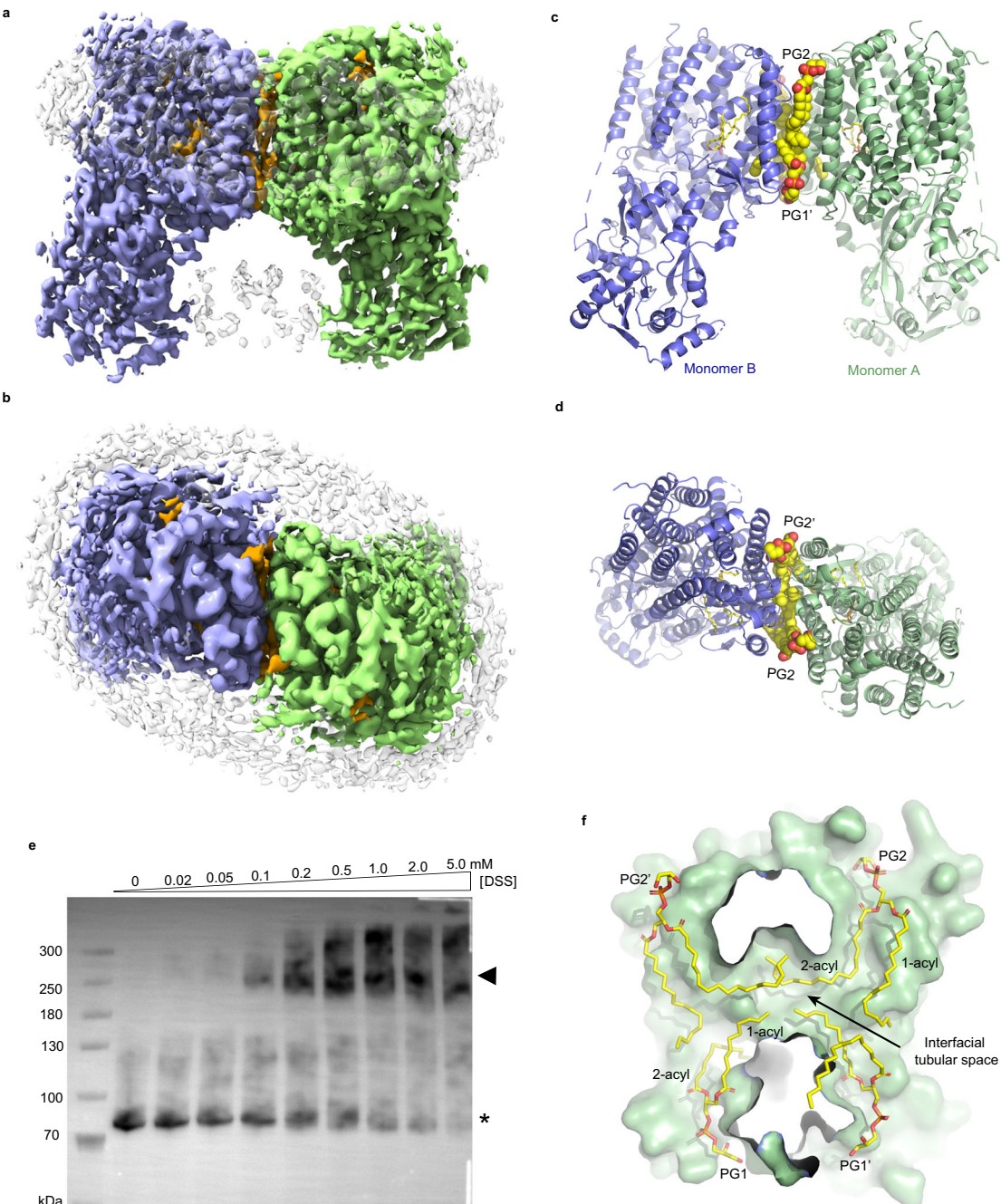

**Fig. 1 Overall structure of *Rt*MprF(DDM)-nanodic homodimer at 3.7-Å resolution.** Cryo-EM densities of *Rt*MprF dimer embedded in a nanodisc viewed along membrane plane (**a**) and along membrane normal from periplasmic side (**b**). Color codes: light green and light blue, two adjacent monomers of *Rt*MprF dimer; yellow, lipid molecules; gray, nanodisc scaffold, uninterpreted lipid and other densities from adjacent *Rt*MprF dimer. Cartoon models of *Rt*MprF dimer viewed along membrane plane (**c**) and along membrane normal from periplasmic side (**d**). The four phospholipid molecules at the dimer interface are highlighted as sphere models and the LysPG inside monomers are shown as stick models. PG phosphatidylglycerol. **e** Western blot of the crosslinked products of *Rt*MprF protein in the membrane. The asterisk indicates the position of *Rt*MprF monomer, while the arrowhead labels the position of *Rt*MprF dimer. DSS was used for the crosslinking experiment. The experiment was repeated independently three times with similar results. **f** The interfacial tubular void space at the dimer interface accommodating the four acyl chains from the PG molecules. The sectional view of the surface model of monomer A is shown and monomer B is omitted for clarity. PG molecules are shown as stick models.

Subdomain 2 (Ser334-Arg530) includes TM9–14. Subdomain 1 harbors a core unit composed of two similar motifs with inverse membrane topology, namely the TM2–TM3a–TM3b–TM4 motif and the TM6–TM7a–TM7b–TM8 motif (Fig. 2c). The two motifs are related to each other through a pseudo-C2 axis running approximately parallel to membrane plane (Supplementary Fig. 5b–d). The overall structure of the *Rt*MprF flippase domain

represents a distinctive membrane protein fold and does not resemble any other lipid transporters with known structures, such as TMEM16F (a lipid scramblase)[30], Lipid II flippase MurJ[31], the ATP-binding cassette transporter MsbA[32], and P4-ATPase lipid-flippase Drsp-Cdc50p[33] (Supplementary Fig. 6).

The carboxyl-proximal region of *Rt*MprF forms a water-soluble synthase domain active in synthesizing aaPG on the cytoplasmic

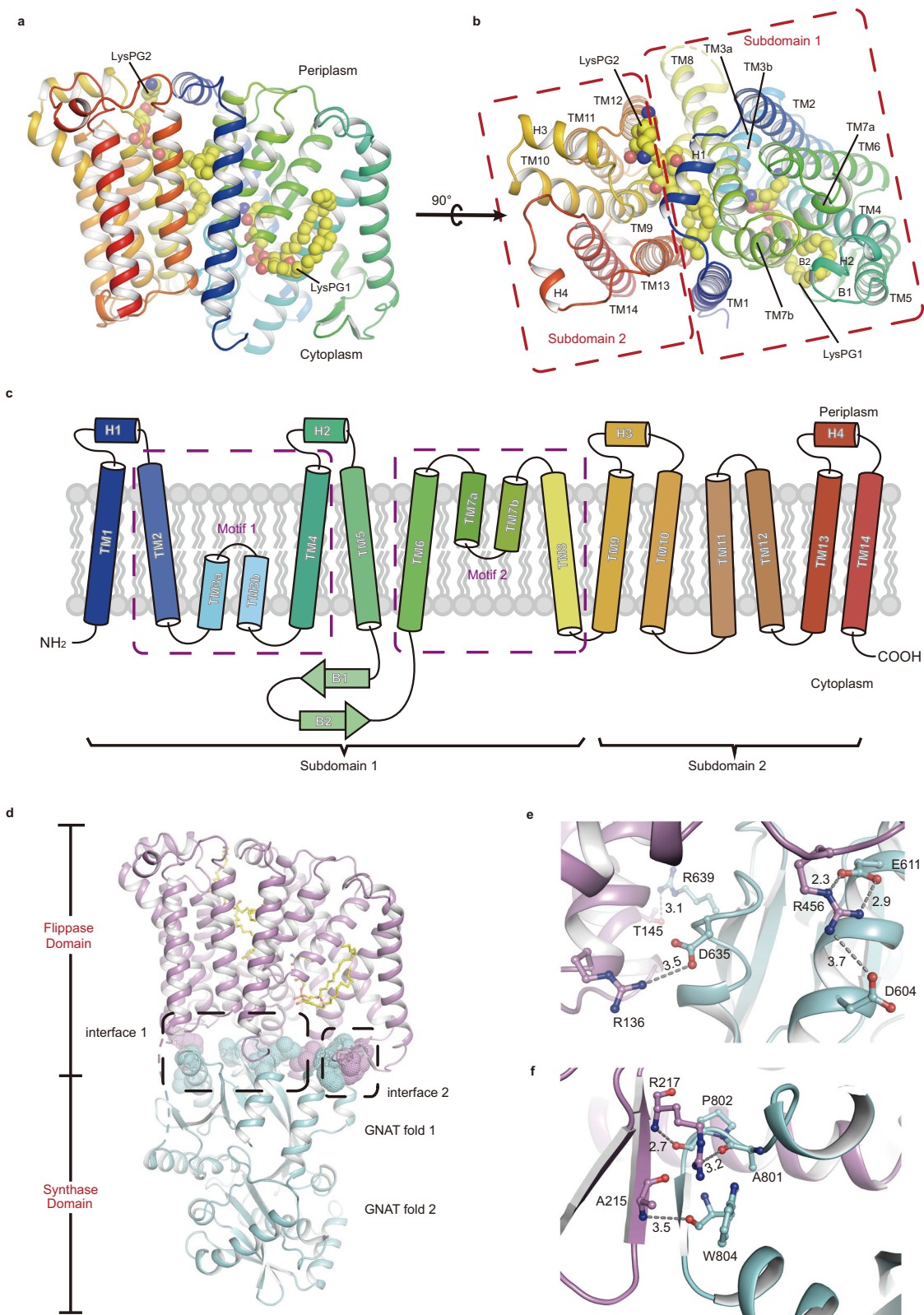

side[8]. It contains the binding sites for aminoacyl-tRNA and PG according to the previous work on the synthase domain structures of MprF homologs from *Pseudomonas aeruginosa* and *Bacillus licheniformis* (*Pa*MprF and *Bl*MprF)[19]. The synthase domain of *Rt*MprF (Pro539-Gly860) superposes well with those of *Pa*MprF and *Bl*MprF (root-mean-square deviation of α-carbons at 0.977 and 1.297 Å, respectively). They share similar

tandem repeats of the General Control Nonderepressible 5 (a histone acetyltransferase of a transcriptional regulatory complex) related N-acetyltransferase (GNAT) folds (GNAT folds 1 and 2) (Supplementary Fig. 4). The GNAT fold 1 of the synthase domain is covalently linked to TM14 in the flippase domain through a flexible loop (residues 531–539, invisible in the map). Meanwhile, it associates closely with the flippase domain through

**Fig. 2 The flippase domain and its interactions with the synthase domain in the *Rt*MprF(GDN)-nanodisc structure solved at 2.96-Å resolution.** A cartoon structural model of *Rt*MprF flippase domain viewed along the membrane plane (**a**) and from periplasmic side (**b**). The endogenous LysPG molecule associated with the flippase domain is highlighted as a sphere model in yellow. TM, transmembrane helix. **c** Topology of the flippase domain of *Rt*MprF protein. The purple dashed boxes indicate the two motifs (Motifs 1 and 2) in Subdomain 1 with inverted topology. H1–H4, amphipathic helices 1–4. B1 and B2, β-strands 1 and 2. **d** Two major contact interfaces between the flippase domain and synthase domain of *Rt*MprF. Color codes: cyan, synthase domain; magenta, flippase domain; yellow, LysPG. The dash boxes indicate the local regions of Interfaces 1 and 2 between the two domains. Zoom-in views of Interfaces 1 (**e**) and 2 (**f**) showing the specific interactions between adjacent amino acid residues from two neighboring domains. The numbers labeled nearby the dash lines are the distances (Å) between two adjacent groups.

non-covalent interactions (Fig. 2d). First, amino acid residues from α3 (Asp604 and Glu611), α4 (Asp635), and β5 (Arg639) regions of the GNAT fold 1 in *Rt*MprF form close interactions (salt bridges and hydrogen bond) with Arg456 from the TM10–TM11 loop, Arg136 from TM3b, and Thr145 from TM4 (Fig. 2e). Their interactions contribute to the formation of Interface 1 between the synthase domain and the flippase domain. Second, the loop region before α10 of GNAT fold 2 contacts with the β-hairpin loop between TM5 and TM6 from the cytoplasmic surface of Subdomain 1 of the flippase domain. This interface (Interface 2) is mainly stabilized by three pairs of hydrogen bonds (Fig. 2f). Among the amino acid residues involved in the inter-domain interactions, Arg456 of *Rt*MprF is highly conserved in other MprF homologs (Supplementary Fig. 7).

**The flippase domain of *Rt*MprF harbors two internal lipid-binding sites within membrane-embedded cavities.** Remarkably, the flippase domain of *Rt*MprF contains two deep cavities located on the cytoplasmic and periplasmic sides of the membrane, respectively (Cavities C and P, Fig. 3a). Cavity C opens to the inner leaflet of the lipid bilayer through a lateral portal measuring 6–8-Å wide (Fig. 3b). Meanwhile, the cavity penetrates deep into a central region of Subdomain 1 near the estimated middle plane of lipid bilayer. The wall of Cavity C is mainly shaped by the transmembrane helices in Subdomain 1, namely TM2, TM3a–3b, TM4, TM5, TM6, TM7a–7b, and TM8. On the other side, Cavity P is surrounded by TM1, H1, TM7b, and TM8 from Subdomain 1 as well as TM9, H3, TM10, TM11, TM12, and TM13 from Subdomain 2. While the internal pocket of Cavity P also extends deep into the central region close to the tip of Cavity C, it has a lateral portal on the other side opening 6–8-Å wide toward the outer leaflet of the lipid bilayer (Fig. 3a). In the central region, the two cavities are separated from each other by a barrier around Arg304 on TM8 (Fig. 3c). Arg304 forms a salt bridge with Glu280 and is hydrogen-bonded to Ala274 and Gly275 from TM7a–7b loop region. The hydrogen bonds between Arg304 and the carbonyl groups of Gly275 and Ala274 might serve to stabilize the side chain of Arg304 in a favorable orientation for establishing ionic interaction with the side chain of Glu280.

Strikingly, one LysPG molecule each (LysPG1 and LysPG2) is trapped in Cavities C and P of the *Rt*MprF(GDN)-nanodisc structure (Fig. 3a). The cryo-EM density of LysPG1 is well-defined and matches well with the model, while the putative LysPG2 shows well-resolved density for the fatty-acyl chains and relatively weak density for the head group. The density of LysPG1 exhibits three arms with similar shape and length (Supplementary Fig. 3b). The first arm is buried inside Cavity C and surrounded by polar amino acid residues, such as Asn117, Asp234, Ser238, and Arg304. The second arm is sandwiched between TM7a and TM4, and surrounded by hydrophobic residues. The density of the third arm is the weakest among the three and it is located at the outmost region exposed to the hydrophobic area of lipid bilayer. Although the local resolution of the three individual arms may appear insufficient for distinguishing the phospho-[3-lysyl

(1-glycerol)] head group and two fatty-acyl groups, interpretation of the lipid molecule is assisted by considering the compatibility of the individual groups with their local environments. As a result, the first arm is assigned as the phospho-[3-lysyl(1-glycerol)] head group and the other two arms most likely belong to the fatty-acyl chains of LysPG molecule. The model is further verified through mutagenesis and biochemical analysis (described below). As shown in Fig. 3b, the LysPG1 molecule in Cavity C has the characteristic hook-shaped polar head group inserted deeply toward the center of Subdomain 1, while the hydrophobic fatty-acyl chains of LysPG1 extend outwardly to the inner leaflet of lipid bilayer. The fatty 2-acyl chains of LysPG1 crawl upward along the external surface of *Rt*MprF and form hydrophobic interactions with non-polar amino acid residues from TM7a and TM4. The head group of LysPG1 bends upward to a position near the middle plane of lipid bilayer, instead of pointing downward to the cytoplasmic surface. In comparison, PG molecules at the dimerization interface adopt the head-group-down inward-facing configuration common to bulk phospholipids in the inner leaflet (Fig. 1f).

The lysyl group of LysPG1 molecule binds to Asp234 and Tyr307 through its side-chain ε-amino group (Fig. 3d). The α-amino group of LysPG is sandwiched between Ala274 and Tyr303, and located merely ~4.4 Å from the barrier site (Arg304) between Cavities C and P. Between the α-amino group of LysPG and Ala274, there is a water molecule serving as a bridge connecting them through hydrogen bonds (Supplementary Fig. 3d). Asp234 of *Rt*MprF is conserved in the homolog from *Pseudomonas aeruginosa* and is replaced by a similar residue (Glu, also an acidic residue favorable for lysyl group binding) in some other species. Besides, Tyr303, Arg304, and Tyr307 in *Rt*MprF are part of the YRXXY motif highly conserved among various MprF homologs (Supplementary Fig. 7). When they are individually mutated to alanine in *Rt*MprF, the amount of LysPG co-purified with the D234A and R304A mutant protein samples is reduced significantly compared to the wild type (Fig. 3e, f), indicating that these two charged residues are crucial for LysPG binding. The results are consistent with the current model of LysPG1 molecule with its lysyl head group buried inside Cavity C. Besides, the head-group glycerol of LysPG1 is hydrogen-bonded to Asn117 and forms van der Waals contact with Phe155. Thereby, the lipid-binding site in Cavity C functions to stabilize the head group of LysPG1 in the upward position through specific interactions. The characteristic hook-like shape of LysPG1 in Cavity C indicates that flipping of LysPG may begin at the initial stage of translocation process on the inner leaflet side instead of the outer leaflet side.

Unlike LysPG1, LysPG2 located in Cavity P has an extended conformation (Supplementary Fig. 3b). Its head group is positioned on the periplasmic surface and both fatty-acyl chains extend deep into the cavity toward the center of Subdomain 1 (Fig. 3a). There are actually three well-resolved fatty-acyl chain densities inside Cavity P of the *Rt*MprF(GDN)-nanodisc structure (Supplementary Fig. 8a). Among the three acyl chains, the two long ones join each other at the head-group region near

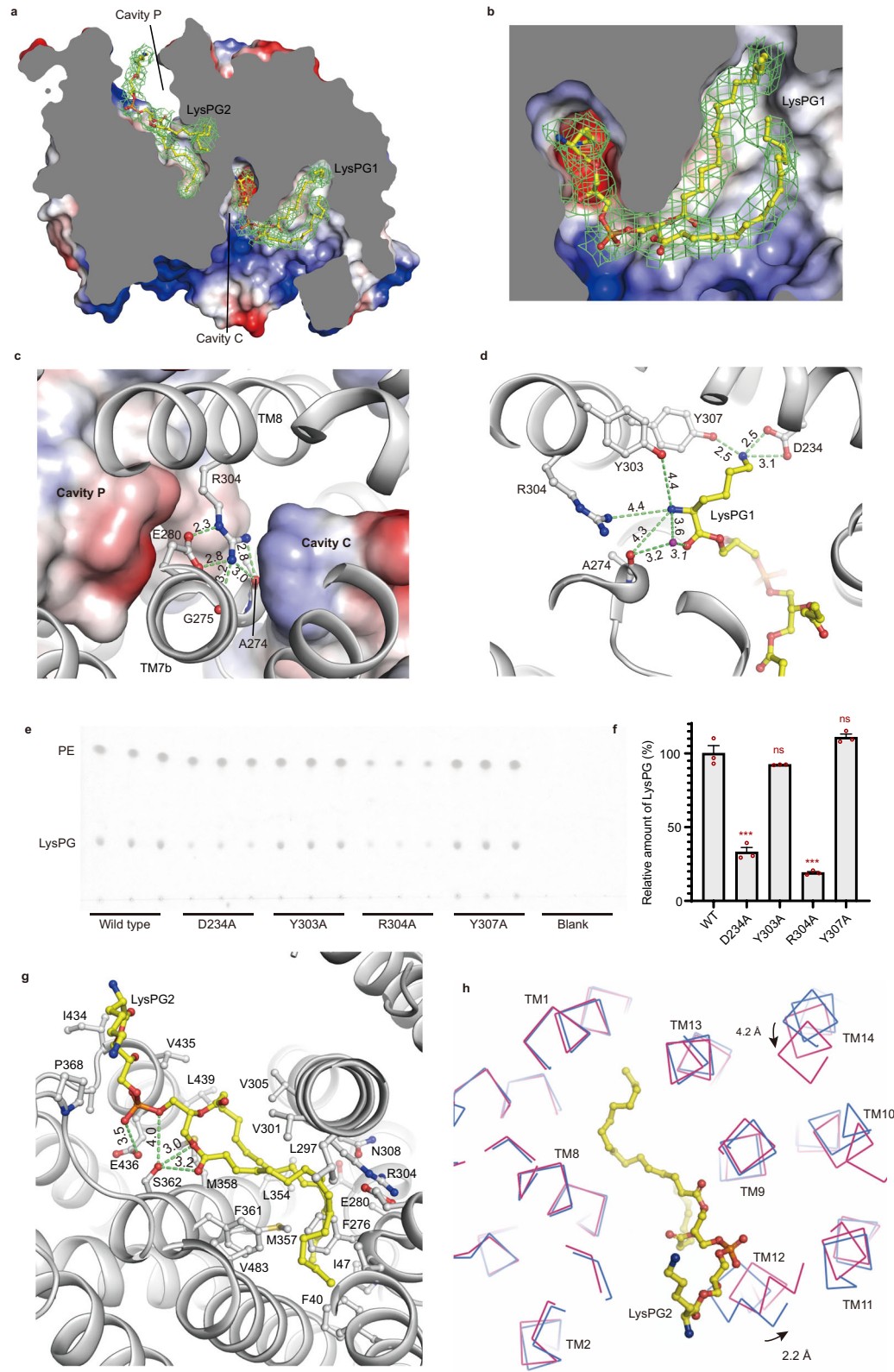

periplasmic surface and belong to a phospholipid molecule tentatively assigned as LysPG2 in the model. The third acyl chain is much shorter than the other two, likely belonging to a detergent molecule. While the density for the two fatty-acyl chains of the phospholipid molecule are fairly strong and clear, the head-group density is relatively weak. When the map contour level is lowered, the density corresponding to the lysyl group becomes visible and

appears to be connected to the glycerol group (Supplementary Fig. 8b). Therefore, the lipid density feature in Cavity P is interpreted as a LysPG molecule with highly flexible lysyl group. Alternatively, a PG molecule may also occupy the site. The backbone glycerol-3-phosphate group of LysPG2 may bind to adjacent residues through hydrogen bonds, whereas the fatty-acyl chains form van der Waals and hydrophobic interactions with

**Fig. 3 Cavities and internal lipid-binding sites in the flippase domain of *Rt*MprF. a** Two intrinsic cavities (Cavities C and P) found in the flippase domain. A cross-sectional view of the electrostatic potential surface model is shown. Blue, positive potential; white, neutral; red, negative potential. The LysPG1 molecule (shown as stick model in yellow) has its head group buried deep in Cavity C. In contrast, the LysPG2 molecule has its head group positioned outside, while two long fatty-acyl chains are buried deep in Cavity P. The cryo-EM densities of the LysPG molecules in Cavities C and P are shown as green meshes (contoured at 3.0 σ level). **b** A zoom-in-view of LysPG1 molecule in Cavity C showing its fatty-acyl chains extending into membrane region through the lateral portal. **c** The contribution of Arg304–Glu280 ionic pair as the barrier between Cavity C and Cavity P. Arg304, Glu280, Ala274, and Gly275 are shown as stick models. **d** Interactions between the head group of LysPG1 and adjacent amino acid residues. **e** Analysis of LysPG co-purified with wild-type *Rt*MprF and mutants through the TLC experiment. The same amount of WT or mutant protein was used for extraction of lipid samples for TLC. The plate was stained by ninhydrin (an amino group-specific dye). PE phosphatidylethanolamine. The lipid spots are identified according to the standard samples of LysPG, PE, and other phospholipids shown in Supplementary Fig. 9d. **f** Quantification of the relative amount of LysPG co-purified with four *Rt*MprF mutants in comparison with the wild type (WT). The error bars indicate the standard errors of the mean values ($n = 3$). $p = 0.000388$ between WT and D234A, ***; $p = 0.2153$ between WT and Y303A, ns not significant; $p = 0.000105$ between WT and R304A, ***; $p = 0.1283$ between WT and Y307A, ns (two-sided unpaired $t$-test between WT and various mutants). The experiment was repeated independently twice with similar results. **g** Interactions of LysPG2 with nearby amino acid residues. The residues in van der Waals contacts or hydrophobic interactions with LysPG2 are shown as silver stick models. The images of **a–d** and **g** represent the structure of *Rt*MprF(GDN)-nanodics at 2.96 Å with two internal LysPG molecules. **h** Superposition of the structures of *Rt*MprF at two different states. Color codes: blue, *Rt*MprF(DDM)-nanodiscs; red, *Rt*MprF(GDN)-nanodiscs. The arrows indicate the putative motion of TM12 and TM14 when the protein switches from one state to the other. In **c**, **d**, and **g**, the numbers labeled nearby the green dashes indicate the distances (Å) between two adjacent groups.

nearby residues including Arg304, Phe276, and several other hydrophobic residues (Fig. 3g). Such a well-resolved lipid feature found in Cavity P is only present in *Rt*MprF(GDN)-nanodisc but not in *Rt*MprF(DDM)-nanodisc.

In the *Rt*MprF(DDM)-nanodisc structure, Cavity C is also occupied by a LysPG molecule similar to the one observed in *Rt*MprF(GDN)-nanodiscs (Supplementary Fig. 3e, b). In contrast, Cavity P only contains some detergent-like density features much weaker than that of LysPG2 in *Rt*MprF(GDN)-nanodiscs. Thin-layer chromatography (TLC) and mass spectrometry analysis results indicate that the *Rt*MprF protein sample does contain LysPG in the lipids co-purified along with the protein (Supplementary Fig. 9a, b). As the *E. coli* cell does not produce endogenous LysPG by itself[6,20], the LysPG molecules bound to *Rt*MprF should be its own product. Through the TLC experiments, the stoichiometry of LysPG co-purified with *Rt*MprF (DDM) protein sample is estimated to be ~1.2 LysPG molecules per *Rt*MprF monomer (Supplementary Fig. 9c). The strong lipid density in Cavity C indicates that it may take up one LysPG molecule, whereas Cavity P in *Rt*MprF(DDM)-nanodiscs is likely occupied by detergent or lipid molecule at very low occupancy. In comparison, the LysPG:protein stoichiometry of the *Rt*MprF (GDN) sample is ~2.6 (Supplementary Fig. 9f, g), much higher than the LysPG:protein stoichiometry of the *Rt*MprF(DDM) sample. Such difference may account for the strong phospholipid density in Cavity P of *Rt*MprF(GDN)-nanodisc due to higher occupancy of LysPG, consistent with the interpretation of the lipid as LysPG2.

Evident conformational differences exist between *Rt*MprF (DDM)-nanodisc and *Rt*MprF(GDN)-nanodisc structures in the regions around Cavity P, despite that their overall structures are similar (Fig. 3h). By superposing them, it is apparent that TM12 moves 2.2 Å closer to TM11 and TM14 moves 4.2 Å closer to TM9 upon binding of LysPG2 in Cavity P. Besides, the amino acid residues involved in binding the fatty-acyl chains of LysPG2 are also adjusted slightly. Previously, it was found that a truncation mutant of *Sa*MprF lacking the bulk region of Subdomain 1 is inefficient in translocating LysPG to the outer leaflet and failed to confer CAMP resistance, whereas the production of LysPG was unaffected[9]. As Subdomain 1 is involved in the formation of lipid-binding sites in both Cavities C and P, the absence of Subdomain 1 will abolish the lipid-translocating function of MprF mainly because the mutant protein cannot either bind LysPG from the inner leaflet or host it on the outer leaflet side.

**The lipid-binding site in Cavity C of *Rt*MprF can accept different aaPGs.** In the flippase domain of *Rt*MprF, the amino acid residues (D234, Y303, R304, Y307) involved in binding the head group of LysPG directly or indirectly are identical or highly similar among various homologs (Supplementary Fig. 7). While MprF homologs from *R. tropici* and *S. aureus* (and many others) catalyze biosynthesis of LysPG, the one from *P. aeruginosa* and one of the two homologs from *C. perfringens* (*Cp*MprF2) produce alanyl-phosphatidylglycerol (AlaPG) instead of LysPG[7,34]. To find out whether the flippase domain of *Rt*MprF can accept AlaPG or not, we have constructed an MprF chimera (*RtPa*MprF) by fusing the flippase domain of *Rt*MprF with the synthase domain of *Pa*MprF (Fig. 4a). The chimeric *RtPa*MprF protein was expressed in *E. coli* and can be purified in sufficient amount for lipid extraction and analysis. The lipid analysis result indicates that the *RtPa*MprF is active in synthesizing AlaPG but not LysPG, and AlaPG could be co-purified along with the *RtPa*MprF protein (Fig. 4b). Mutation of the four key residues involved in binding LysPG results in significant decrease of the amount of AlaPG co-purified with the protein (Fig. 4c). Therefore, it is apparent that the preference for different aaPG is mainly conferred by the synthase domain of *Rt*MprF, presumably by the selective binding of different aminoacyl-tRNA in its active site.

For the flippase domain of *Rt*MprF, the substrate specificity appears to be broad, allowing it to accept either LysPG or AlaPG. Similarly, the flippase domains of MprF homologs from *S. aureus* and *C. perfringens* also exhibit relaxed substrate specificities[35]. The head groups of LysPG and AlaPG most likely share the same binding pocket in Cavity C of the flippase domain. Binding of AlaPG in the pocket is more sensitive (than LysPG) to the mutation of the four key residues involved in aaPG binding, as the alanyl group contains a small side chain and may require the presence of the bulky side chains of Tyr303 and Tyr307 in the flippase domain of *Rt*MprF for its binding through van der Waals interactions.

**The LysPG-binding site in Cavity C is involved in regulation of LysPG synthesis and translocation.** What is the functional role of the LysPG-binding site in Cavity C of *Rt*MprF protein? Do the mutations of LysPG-binding sites affect the overall level of LysPG production and the flippase function under in vivo condition? To answer the questions, we have analyzed the overall LysPG levels of *E. coli* cells expressing D234A, Y303A, R304A, and Y307A mutants of *Rt*MprF and compared them to the cells expressing wild-type protein. Remarkably, the cells expressing the mutant

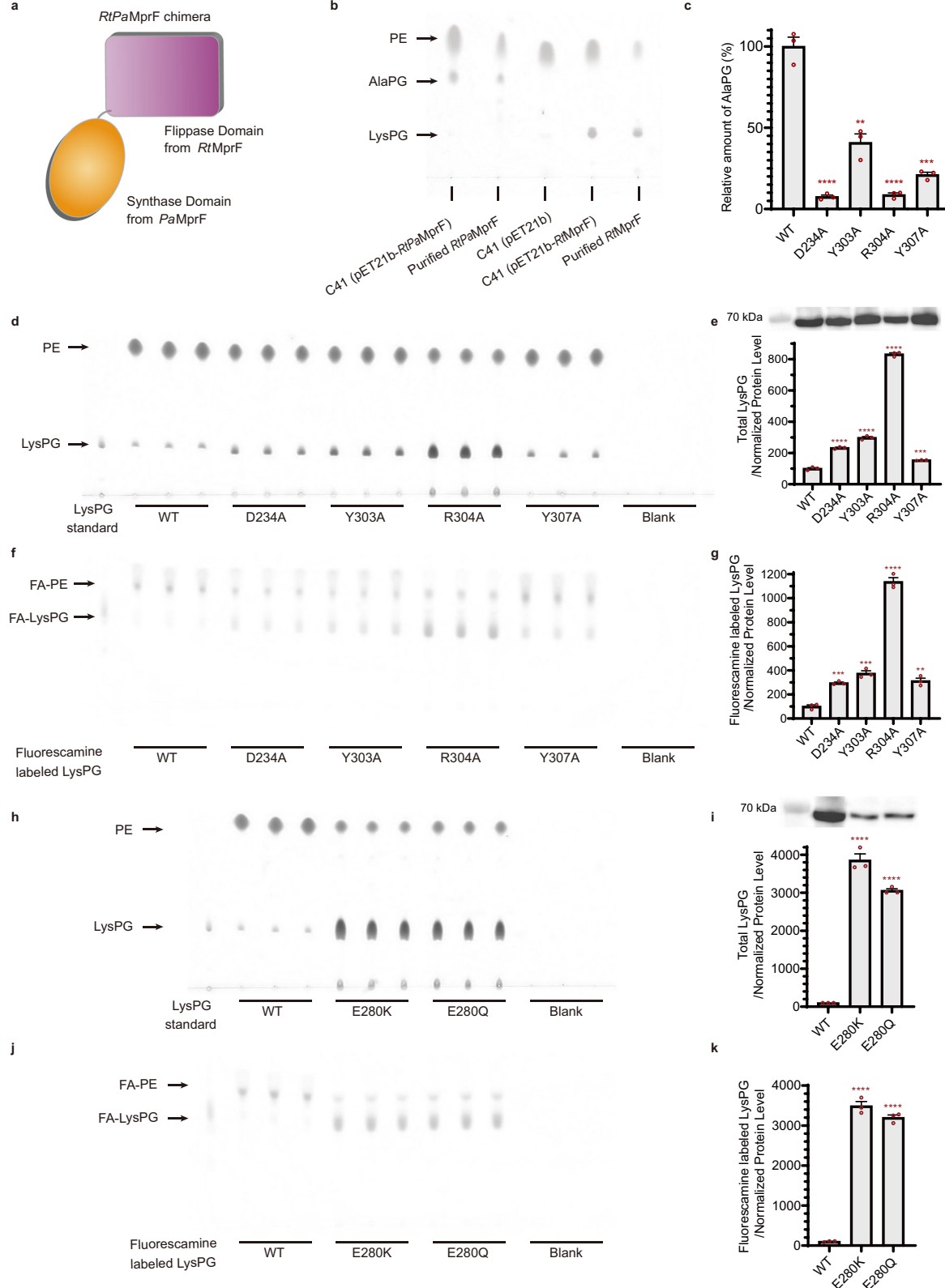

proteins have higher level of LysPG (per unit protein) than those with the wild-type protein (Fig. 4d, e), while the protein expression levels are slightly variable among the wild type and mutants (Fig. 4e). Among the four mutants, the cells expressing R304A mutant exhibit the highest level of total LysPG, while those with Y307A mutant have lower level of total LysPG than the other three mutants but are still 1.5 times of those with wild-type

protein. Similarly, the levels of LysPG on the outer leaflet of membrane (accessible by the fluorescamine dye) appear to be also higher for the mutants than for the wild type (Fig. 4f, g). Fluorescamine is a membrane-impermeable dye mainly reacting with the LysPG exposed on cell surface (on the outer leaflet) and was previously used for characterizing the in vivo flippase function of *Sa*MprF in *S. aureus*[9].

**Fig. 4 Substrate selectivity and functional role of the LysPG-binding site in Cavity C of *Rt*MprF. a** Cartoon diagram of the *RtPa*MprF chimera constructed by fusing the flippase domain from *Rt*MprF (residues 1–541, violet) and the synthase domain from *Pa*MprF (residues 554–881, orange). **b** Analysis of the lipid samples extracted from the cell expressing the *RtPa*MprF chimera and purified protein samples through the TLC method. The lipid extracted from cells carrying empty vector (pET21b) and pET21b-*Rt*MprF construct is loaded as controls. The lipid spots on the TLC plate were stained by ninhydrin. **c** Relative amount of AlaPG co-purified with *RtPa*MprF protein and various mutants. $p = 0.000093$ between WT and D234A, ****; $p = 0.001696$ between WT and Y303A, **; $p = 0.0001$ between WT and R304A, ****; $p = 0.00019$ between WT and Y307A, ***. **d** TLC result showing the overall LysPG content extracted from *E. coli* cells expressing wild-type and various mutants of *Rt*MprF. The positions of LysPG spots on the TLC plate are indicated by LysPG standard loaded on the left in parallel. **e** The total LysPG content per arbitrary unit of protein level. The data are based on the lipid spot intensity on the TLC plate (**d**) divided by the normalized protein level shown above the plot. $p = 0.000049$ between WT and D234A, ****; $p = 0.00003$ between WT and Y303A, ****; $p = 2.4e − 7$ between WT and R304A, ****; $p = 0.001$ between WT and Y307A, ***. **f** TLC result of the fluorescamine-labeled LysPG from *E. coli* cells expressing wild-type and various mutant *Rt*MprF proteins. FA-PE fluorescamine-labeled PE, FA-LysPG fluorescamine-labeled LysPG. **g** Quantification of the relative fluorescamine-labeled LysPG content per arbitrary unit of protein level. The data are based on the lipid spot intensity on the TLC plate (**f**) divided by the normalized protein level shown above the plot in (**e**). The positions of fuorescamine labeled LysPG spots on the TLC plate are indicated by the LysPG standards labeled with fuorescamine (loaded on the left side). $p = 0.000237$ between WT and D234A, ***; $p = 0.000406$ between WT and Y303A, ***; $p = 0.000009$ between WT and R304A, ****; $p = 0.0013$ between WT and Y307A, **. **h** TLC result showing the overall content of LysPG extracted from *E. coli* cells expressing E280K and E280Q mutants in comparison with the wild type. **i** Quantification of the relative LysPG content for cells hosting E280K and E280Q mutants per arbitrary unit of protein level. The data are based on the lipid spot intensity on the TLC plate (**d**) divided by the normalized protein level shown above the plot. $p = 0.000024$ between WT and E280K, ****; $p = 4.3e − 7$ between WT and E280Q, ****. **j** TLC analysis on the fluorescamine-labeled LysPG content in *E. coli* cells expressing E280K and E280Q mutants in comparison with the wild type. **k** Quantification of the relative fluorescamine-labeled LysPG content per arbitrary unit of protein level for E280K and E280Q mutants. $p = 0.000007$ between WT and E280K, ****; $p = 0.000001$ between WT and E280Q, ****. The data are based on the lipid spot intensity on the TLC plate (**j**) divided by the normalized protein level shown above the plot in (**i**). The protein bands shown above the bars in (**e**) and (**i**) are western blot bands (with anti-His tag antibody) of the cells (loaded with same amount for wild type and different mutants) expressing target proteins. The identities of western blot bands are the same as those labeled below each individual bars. In **e**, **g**, **i**, and **k**, total or fluorescamine-labeled LysPG/normalized protein level $= 100 \times [(L_{mutant} − L_{blank})/(L_{wt} − L_{blank})]/[(P_{mutant} − P_{blank})/(P_{wt} − P_{blank})]$. $L_{mutant}$ and $L_{wt}$ represent the intensities of the designated lipid spots from *E. coli* cells expressing mutant and wild-type *Rt*MprF proteins, respectively, while $L_{blank}$ represents the background intensity. $P_{mutant}$ and $P_{wt}$ represent the intensities of western blot bands for mutant and wild-type *Rt*MprF proteins, respectively, while $P_{blank}$ represents the background intensity. The error bars in (**c**, **e**, **g**, **i**, and **k**) indicate the standard errors of the mean values ($n = 3$). **$p < 0.01$, ***$p < 0.001$, ****$p < 0.0001$ (two-sided unpaired *t*-test between WT and various mutants). For **b**, **d**, **f**, **h**, and **j**, each experiment was repeated independently twice with similar results.

To further analyze the functional role of Glu280 in *Rt*MprF, three point mutants, namely E280A, E280K, and E280Q, were generated. While the protein expression level of E280A mutant is too low to be detected in western blot, the E280K and E280Q mutants can be expressed, but at much lower level than the wild type (Fig. 4i). Nevertheless, the E280K and E280Q mutants exhibit more than 30 times increase in the total LysPG level per unit protein relative to the wild type, despite that the two mutant proteins express at much lower levels than the wild type (Fig. 4h, i). Similarly, the amounts of fluorescamine-labeled LysPG per unit protein also increase significantly in the two mutants (Fig. 4j, k).

Moreover, the proportion of LysPG in total phospholipids extracted from cells expressing the six mutants mentioned above are about 1.5–5 times of the LysPG/total phospholipids ratio in cells expressing wild-type *Rt*MprF (Supplementary Fig. 9d, e). Therefore, the mutations on the LysPG-binding sites in Cavity C or the Glu280–Arg304 ionic pair of *Rt*MprF have dramatic stimulating effect on its synthase function, presumably by removing the potential inhibitory effect imposed by LysPG bound to Cavity C. Besides, the mutations may also enhance its flippase function leading to increased level of LysPG on the outer leaflet of the membrane.

**_Rt_MprF exhibits variable conformations with the synthase domain rearranged relative to the flippase domain.** MprFs utilize aminoacyl-tRNA from cytosol and PG from membrane as the substrates for aminoacyl phospholipid synthesis[7]. The potential binding site of lysyl-tRNA in the synthase domain of *Rt*MprF is located at the cleft between GNAT folds 1 and 2 (Supplementary Fig. 10a, b), according to the previous studies on an alanyl transferase FemX involved in peptidoglycan biosynthesis (in complex with an aminoacyl-tRNA analog) and *Bl*MprF in complex with L-lysine amide (LYN)[19,36]. Apparently, the active site of the synthase domain is separated from the LysPG-binding sites in the flippase domain by a large distance at 47.0 (within the same subunit) or 71.4 Å (between adjacent subunits). Such a large gap is apparently unfavorable for direct translocation of lipid molecules (PG or LysPG) between the two domains. To overcome the problem, *Rt*MprF may go through large conformational changes and rearrange the synthase domain to a position close to membrane surface in order to acquire PG from the membrane and release LysPG back to the membrane.

In addition to the major class (class A) of symmetrical *Rt*MprF dimer, there are minor classes (classes B–D) exhibiting asymmetric conformations (Supplementary Fig. 10c). Within the asymmetric *Rt*MprF dimers, one monomer rearranges the synthase domain more dramatically than the adjacent one. When the flippase domains of classes B–D are superposed with that of class-A symmetric dimer, it becomes apparent that the synthase domains adopt distinct positions in the minor classes (Supplementary Fig. 10d–f). For class A, the long axis of the synthase domain forms 115.2° angle with the membrane plane. In comparison, the corresponding ones of classes B–D form much smaller angles (12.8–32.4°) with the membrane plane (Supplementary Fig. 10c). In these cases, the synthase domains may rotate from the upright position to nearly horizontal positions and detach from the cytoplasmic surfaces of the flippase domains. Consequently, the restraints restricting its movement are greatly reduced so that the synthase domain can readily move across long distance through Brownian motion. The inter-domain flexibility of *Rt*MprF, as reflected by the variable positions of the synthase domain, may allow it to change conformation so as to acquire hydrophobic substrate (PG) from the membrane and to release LysPG back to the membrane for translocation by the flippase domain.

## Discussion

Single-nucleotide polymorphisms of *Sa*MprF are frequently found to be associated with daptomycin resistance of *S. aureus*,

whereas the action mechanism of *Sa*MprF in causing daptomycin resistance remains largely unclear[11,13]. A recent work reported that T345A single-nucleotide polymorphism of *Sa*MprF can cause daptomycin resistance in *S. aureus* reproducibly, while the mutation did not affect LysPG synthesis or translocation[14]. To analyze the locations of DAP-R-related mutations on *Sa*MprF[14,37], a structural model of *Sa*MprF is generated through the comparative protein structure modeling method[38]. The mutation sites are mainly located in the flippase domain, while only three of them are in the synthase domain (Supplementary Fig. 11a, b). Among the 24 sites in the flippase domain of *Sa*MprF, eighteen are located in Subdomain 2 and the remaining six are in Subdomain 1. Interestingly, TM9 in Subdomain 2 contains a region of high-frequency DAP-R mutations including two gain-of-function mutations (T345A and V351E) responsible for significantly enhanced DAP-R phenotype[14] (Supplementary Fig. 11c). It was proposed that the mutations may modulate specific interactions of *Sa*MprF with the antibiotic molecule instead of affecting LysPG synthesis or translocation[14].

Computational docking analysis suggests that daptomycin molecule can fit well in Cavity P of *Sa*MprF (Supplementary Fig. 11d). The daptomycin molecule inserts its N-terminal decanoyl fatty-acyl group and tryptophan side chain in two deep pockets of Cavity P (Supplementary Fig. 11e). Several DAP-R mutations are located near the entrance of Cavity P (Supplementary Fig. 11f) or on the wall of its internal pockets (Supplementary Fig. 11g). The mutations may influence the interactions between daptomycin and *Sa*MprF by altering the overall shape and surface property of Cavity P. While the earlier work[14] and our hypothetical model both suggest that daptomycin may interact with *Sa*MprF, more biochemical and other evidences are needed to verify their interaction and to address the question about whether *Sa*MprF can directly translocate daptomycin or not. The structural model serves as a preliminary framework to guide further experiments in order to reveal the mechanism of daptomycin resistance caused by *Sa*MprF variants. Moreover, the model might also be useful for discovery and development of *Sa*MprF inhibitors as anti-infective agents. In practice, the *Sa*MprF model could potentially be used in the structure-based virtual screening of antibiotic drugs for treating MRSA or VRSA infections.

As shown in the structure and crosslinking experiment (Fig. 1), *Rt*MprF protein forms homodimer and larger oligomers, consistent with previous biochemical study on *Sa*MprF[26]. There might be potential functional advantages for the *Rt*MprF proteins to form homo-oligomers (dimer or tetramer). First, the dimerization interface of *Rt*MprF homodimer contains four PG molecules, which may also serve as substrate for the synthase domain. Upon dimerization, the lipid substrate (PG) of *Rt*MprF may be enriched in the milieu to promote synthesis of aaPG. Second, the two monomers may help each other in translocating aaPG if it is released to the membrane after being produced by the synthase domain. Dimerization or tetramerization of MprF may help to concentrate multiple flippase domains in a local region enriched with LysPG so that the translocation of LysPG can occur more efficiently as free diffusion of LysPG in the membrane is a relatively slow process and translocation of lipid molecule across the membrane is also a rate-limiting process[39].

In bacterial cells, MprFs carry out dual functions by synthesizing aaPG and translocating it to the outer leaflet of bacterial membrane[9]. Although the synthase domain of MprF alone can catalyze synthesis of LysPG by itself under in vitro conditions, efficient production of LysPG in vivo requires the presence of both domains, and it was suggested that the transmembrane domain of MprF may help to position the catalytic domain in the cytosol during aaPG synthesis[8]. The cryo-EM structure of

*Rt*MprF in complex with LysPG reveals the close relationship between the two domains and apparently, the membrane-embedded flippase domain does serve to position the catalytic domain in the cytosol by forming specific interactions with it (Fig. 2d). As shown in Fig. 2e, f, the catalytic domain interacts with amino acid residues from the cytoplasmic surfaces of Subdomains 2 and 1 mainly through salt bridges and hydrogen bonds. Such specific interactions enable the catalytic domain to approach membrane surface and acquire the lipid substrate (PG) from the membrane more efficiently than the catalytic domain expressed alone (not attached to the Subdomain 2). Although the in vitro activity assay reveals that the purified catalytic domain of MprF is active in the absence of transmembrane domain[8,19], the function of isolated catalytic domain of MprF relies on delivering of lipid substrate through detergent (Triton X-100) micelles in the solution, a condition which does not exist in the in vivo environments. Therefore, the catalytic domain expressed alone in *E. coli* is not functional in producing of aaPG[8], mainly because it is inefficient in acquiring the lipid substrate from the membrane through random diffusion process.

The characteristic hook-shaped LysPG1 and its binding site in Cavity C observed in the structures shed light on the mechanism of lipid selectivity in the flippase domain of MprF. The aminoacyl head groups of LysPG or AlaPG are specifically recognized by amino acid residues on the wall of Cavity C as observed in the *Rt*MprF structure (Fig. 3d). The electronegative surface within Cavity C (Fig. 3b) is favorable for the binding of positively charged aminoacyl group of LysPG or AlaPG, but unfavorable for the anionic phospholipids (such as PG and cardiolipin (CL)) to bind. Moreover, the head groups of phosphatidylethanolamine (PE), phosphatidylserine (PS), and PG are much smaller in size and lacks the lysyl or alanyl group when compared with LysPG and AlaPG. Even if PE, PS, or PG from the membrane can enter Cavity C in MprF, they cannot form stable interactions with nearby residues. As for CL, it contains four fatty-acyl chains and is too big to fit in Cavity C of MprF. Therefore, the unique phospholipid-binding site in Cavity C of MprF likely selects LysPG or AlaPG against the other phospholipids through compatible shape, size, and surface charge property.

Near the interface between the two domains of *Rt*MprF, LysPG is found within the inner cavity (Cavity C) of the flippase domain instead of binding to the active site of the synthase domain. Therefore, the *Rt*MprF(DDM)-nanodisc structure may represent an intermediate state of the *Rt*MprF protein after LysPG is synthesized and before it is translocated to the outer leaflet. How does MprF coordinate the two domains to facilitate synthesis of LysPG at the interface between cytosol and membrane surface (on the intracellular side) under physiological conditions? After LysPG is synthesized, how is it translocated by MprF from the inner leaflet to the outer leaflet of the membrane? To address the questions, here we propose a mechanistic model to account for the LysPG synthesis and translocation process mediated by MprF basing on the structural and biochemical analysis results of *Rt*MprF (Fig. 5). For MprF to acquire the substrate PG molecules from membrane, the synthase domain may need to approach the cytoplasmic surface of the membrane in a nearly horizontal position (State 1). After LysPG is synthesized and tRNA is released, the synthase domain returns to the upright position (State 2) and LysPG in the inner leaflet of membrane diffuses into the binding site in Cavity C (State 3, as observed in *Rt*MprF (DDM)-nanodisc structure with one LysPG bound). Diffusion of LysPG into Cavity C and its translocation is likely driven by the electrostatic repulsive force from the positively charged cytoplasmic surface near the entrance of Cavity C. Subsequently, the gate around the Arg304–Glu280 ionic pair might open transiently so that LysPG can migrate from Cavity C to Cavity P through a

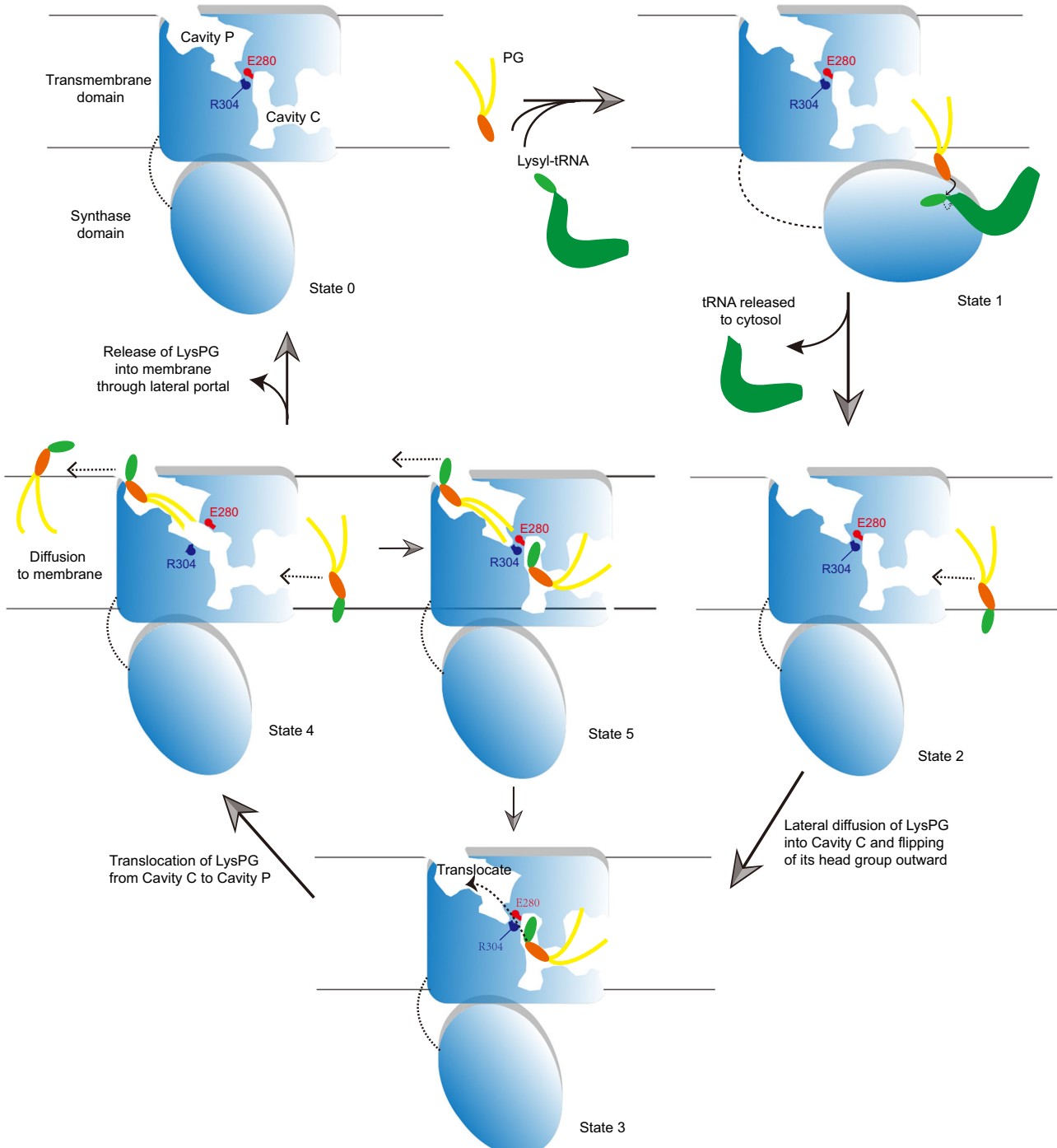

**Fig. 5 A mechanistic model of LysPG synthesis and translocation mediated by MprF.** While each monomer in the homodimer may mediate the entire cycle of LysPG synthesis, translocation and release independently, the other monomer may also serve to stabilize the adjacent one in the optimal position (relative to lipid bilayer) for synthesis and translocation of LysPG. The cavities and Arg304–Glu280 pair are only shown in the active monomer and the other monomer is left out for clarity. State 0 represents a resting state of MprF. At State 1, the curved solid arrow indicates the nucleophilic attack of the PG head-group hydroxyl on the activated α-carbon of lysyl-tRNA and the dash arrow denotes the break of covalent bond between lysyl group and tRNA. At States 2 and 4, the dash arrows indicate the lateral diffusion of LysPG into and out of the cavities in *Rt*MprF. Loading of LysPG into Cavity C leads to State 3 as observed in the *Rt*MprF(DDM)-nanodisc structure. During the transition from State 3 to State 4, a channel connecting Cavity C and Cavity P may emerge transiently to facilitate translocation of LysPG. When the Cavity C in State 4 is refilled with a second LysPG molecule, both cavities are occupied by LysPG as observed in the *Rt*MprF(GDN)-nanodisc structure (State 5). Further releasing of LysPG from Cavity P to the membrane leads to switching of State 5 to State 3.

putative channel connecting the two cavities (State 4). A second molecule of LysPG may further enter Cavity C, so that Cavities C and P are both loaded with LysPG as observed in the *Rt*MprF (GDN)-nanodisc structure (State 5). On the outer leaflet side, the LysPG molecule in Cavity P might be attracted by the negatively charged periplasmic surface around the lateral portal so that it can be released to the outer leaflet of the membrane. The LysPG translocation process may cycle among States 3–5 till the LysPG pool in the inner leaflet side is depleted. Afterwards, MprF returns to the initial apo-state (State 0) for the next round of LysPG synthesis-translocation process.

In the *Rt*MprF(GDN)-nanodisc structure, there is an additional lipid-like density nearby LysPG1 (Supplementary Fig. 12). The density is sandwiched by LysPG1 and the β-hairpin loop between TM5 and TM6. As the molecule is located at the entrance of Cavity C, it may serve as a potential site for LysPG loading after it is synthesized and released from the synthase domain. Alternatively, in case that a PG molecule is captured at this site, it may serve as the substrate for the synthase domain if it could diffuse laterally to the active site of the synthase domain at a horizontal position close to the membrane surface. The third possibility is that it may belong to the bulk lipid (such as PE or elses) from the membrane, serving to stabilize LysPG1 in Cavity C and prevent unloading of LysPG1 on the cytoplasmic side by blocking the cytoplasmic portal of Cavity C.

While the head group of LysPG1 is blocked from entering Cavity P by the putative gate around the Arg304–Glu280 ionic pair in *Rt*MprF (Figs. 3c and 5, State 3), the remaining parts of LysPG1 (including the glycerol-3-phosphate backbone, two fatty-acyl chains and the head-group glycerol group) are prevented from entering Cavity P by the steric hindrance effects of amino acid residues from TM7a, TM7b, TM3a, TM3b, as well as the TM7a–TM7b and TM3a–TM3b loops. Among them, His271 on TM7a–7b loop region serves to block the 1-acyl chain of LysPG from entering the cleft between the TM7a–TM7b and TM3a–TM3b loops. Such a narrow cleft is closed at a contact point where Pro273 and Ala274 (in the TM7a–TM7b loop region) form van der Waals interaction and hydrogen bond with Phe121 and Thr118 in the TM3a–TM3b loop region, respectively. Pro273 of *Rt*MprF is highly conserved in other MprF homologs (Supplementary Fig. 7) and it corresponds to Pro247 in *Sa*MprF. The P247A mutant of *Sa*MprF not only affects Lys-PG production, but also increases daptomycin susceptibility of *S. aureus*, suggesting a decreased flippase activity of the mutant compared to the wild type[26]. For the PG group of LysPG1 to pass through the cleft and enter Cavity P, either TM7a–TM7b motif or TM3a–TM3b motif needs to move away from the central region so that a larger portal can form between them so as to facilitate translocation of LysPG1 into Cavity P.

Expression of a construct with Subdomain 1 (residues 1–320) of *Sa*MprF separately can support LysPG translocation to some degree, when the synthase domain (328–840, including Subdomain 2 and the C-terminal catalytic domain) is co-expressed along with the Subdomain 1 in the same cell[26]. Through a bacterial two-hybrid assay, Ernst et al. discovered that Subdomain 1 interacts with Subdomain 2 (also termed Syn-h in the literature) and the C-terminal catalytic domain (Syn-cyt)[26], and these interactions have been unraveled in detail through our structural analysis on *Rt*MprF (Fig. 2d, e). It was also demonstrated that Subdomain 1 is essential for the flippase activity of *Sa*MprF and the optimal flippase function may rely on the interaction of Subdomain 1 with the synthase domain (Syn-h-cyt)[14,26]. As shown in the structure of *Rt*MprF (Fig. 3), Subdomain 1 harbors most of the LysPG-binding site in Cavity C and part of Cavity P. As Cavity P may serve to host LysPG temporarily and facilitate release of LysPG to the outer leaflet as proposed in our

mechanistic model, the integrity of its local structure is probably essential for the optimal flippase activity of MprF. Consistently, when Subdomain 1 was fused with two extra transmembrane helices from Subdomain 2 (or Syn-h) and co-expressed along with the synthase domain, it exhibited higher flippase activity than the one without the extra transmembrane helices[26].

Does the aaPG translocation process mediated by MprF requires energy? Unlike the P-type ATPase lipid flippase (Drs2p-Cdc50p) or ATP-Binding Cassette transporter (MsbA), MprF does not contain any ATPase or ATP-binding cassette domain (Supplementary Fig. 6). It is not a primary active transporter and does not utilize the energy of ATP to transport aaPG. Recently, a bioinformatics study suggests that MprF may belong to the Major Facilitator Superfamily (MFS) basing on the observation that the transmembrane region of MprF contains a MFS-like domain[40]. The MFS members mediate facilitated diffusion or cation (H+ or Na+)-dependent transport and solute:solute antiport of substrate molecules[41]. For the proton-dependent oligopeptide transporters of the MFS superfamily, they share a conserved ExxER/K motif in the transmembrane domain and two pair of salt bridges serving to stabilize the transporter at different conformational states[42]. In *Rt*MprF, only one pair of salt bridge is found in the membrane-embedded region, namely the Glu280–Arg304 pair. In the previous work by Ernst et al.[26], an alanine mutation was introduced to the Asp254 site of *Sa*MprF (corresponding to Glu280 in *Rt*MprF), and the *S. aureus* strains with the D254A mutant exhibited significantly increased daptomycin susceptibility (suggesting a reduced flippase activity of *Sa*MprF) compared to the wild type. Moreover, the R279A mutation (corresponding to Arg304 in *Rt*MprF) also leads to similar loss-of-function effect. For *Rt*MprF, while the E280A mutant does not express in *E. coli* (presumably due to high toxicity of the target protein), the E280K and E280Q mutants (Fig. 4h–k) as well as the R304A mutant (Fig. 4d–g) can be expressed in *E. coli* and exhibit enhanced level of both synthase and flippase activities relative to the wild type. The distinct functional phenotypes between R304A of *Rt*MprF and R279A of *Sa*MprF may be due to the differences in protein expression level and host species.

As the Glu280–Arg304 pair is located at the gate between the two cavities, protonation of Glu280 may lead to weakening of Glu280–Arg304 interaction and trigger further conformational changes nearby to open the gate and allow the substrate (LysPG) to go through. In this case, the proton gradient across the membrane might be exploited by MprF to stimulate or drive the translocation of LysPG molecule.

As LysPG is initially synthesized on the cytoplasmic side and inserted to the inner leaflet of the membrane, the concentration of LysPG should be higher in the inner leaflet side than [LysPG] in the outer leaflet. Moreover, there is a LysPG hydrolase found in some bacterial periplasma serving to hydrolyze LysPG and lower its concentration in the outer leaflet constantly[43,44]. Therefore, there should be a concentration gradient of LysPG between the inner leaflet and the outer leaflet of the membrane at the initial stage. In this case, MprF likely serves as a uniporter to facilitate diffusion of LysPG down its concentration gradient and the process might not need to consume energy, but it could probably be stimulated by protonation of Glu280. Alternatively, it may function like a lipid scramblase, such as TMEM16, which harvest the energy of the phospholipid gradient and may utilize a hydrophilic cavity on the surface of membrane-embedded domain for transport of lipids[45]. For future works, analyzing the flippase activity of MprF through the liposome-based transport assay under asymmetric (and symmetric) pH conditions across the membrane will be helpful in addressing the question about whether the flippase activity of MprF is regulated by proton/ion gradient or not.

## Methods

**Protein expression and purification.** The *mprF* gene from *R. tropici* (*Rt*MprF) was cloned into the pET21b vector and the recombinant plasmid was used to transform *E. coli* C41(DE3) cells. The cells were cultured in Terrific Broth media containing 50-μg mL⁻¹ ampicillin, and protein expression was induced with 0.5-mM isopropyl β-D-thiogalactopyranoside (IPTG) overnight at 16 °C after OD$_{600}$ reached 1.0. For purification of *Rt*MprF, cells were harvested through centrifugation, resuspended in a buffer containing 25-mM Tris-HCl (pH 8.0) and 300-mM NaCl and then incubated with 1% lysozyme at 4 °C for 30 min. The suspension was sonicated and centrifuged at 11,000 $g$ (JL-25.50 rotor, Beckman) for 30 min. The supernatant was ultracentrifuged at 158,000 $g$ (Type 45 Ti rotor, Beckman) to collect membrane pellets. The pellets were resuspended in a buffer with 25-mM Tris-HCl (pH 8.0), 300-mM NaCl, and 1.5% β-DDM (Anatrace) and incubated at 4 °C for 30 min. After centrifugation at 40,000 $g$ for 30 min, the supernatant was loaded onto a Ni-NTA column and the protein was purified at 4 °C by a step-wise elution method with three different buffers containing 25-mM Tris-HCl (pH 7.5), 300-mM NaCl, 20, 50, or 300-mM imidazole, and 0.05% β-DDM for MprF(DDM) sample or 0.02% GDN for MprF(GDN) sample. The fraction eluted in the buffer with 300-mM imidazole was pooled and concentrated to 10 mg mL⁻¹ in 100-kDa molecular weight cutoff (MWCO) concentrator (Millipore) for further experiments.

To construct the *RtPa*MprF chimera, the cDNA sequence of *Rt*MprF synthase domain (residues 542–869) on the pET21b-*Rt*MprF vector was replaced by the coding sequence of *Pa*MprF synthase domain (residues 554–881, AlaPG synthase) through two PCR reactions. In detail, the region encoding *Pa*MprF synthase domain was amplified through the first PCR reaction by using a pair of primers with sequences of 5′-CCGGCAACGAAGCGGCCCGGAGCCTGTCAGCGCGGAA GAGCTG-3′ and 5′-GTGGTGGTGCTCGAGTGCGGCCGCAAGCTTGCGTTTC ACCAA-3′. The DNA product contains 30-bp nucleotides in the 5′- and 3′-terminal regions matching with the regions upstream and downstream of the coding sequence of *Rt*MprF synthase domain on the vector pET21b, respectively. The replacement was further accomplished through a second PCR reaction. by using the DNA product of the first round PCR (with the cDNA encoding *Pa*MprF synthase domain) as primers and the pET21b-*Rt*MprF vector as template. The second PCR reaction adopts the protocol of the Quick Change method. After digestion of the template with DpnI, the product of the second round of PCR was used for transformation of DH5α *E. coli* competent cells for plasmid amplification. After transformation and antibiotics resistance screening, the clone with target plasmid was selected and verified through DNA sequencing. The protocols for *RtPa*MprF protein expression and purification were the same as the one used for wild-type *Rt*MprF.

The DNA encoding the synthase domain of *Rt*MprF (*Rt*MprF-SD, residues 542–862) was cloned into the pET21b vector and transformed into the *E. coli* BL21 (DE3) cells for protein expression. For purification of the recombinant *Rt*MprF-SD protein expressed in BL21(DE3) cells, the cell pellets harvested through centrifugation were resuspended in a buffer with 25-mM Tris-HCl (pH 8.0) and 700-mM NaCl. After the cells were lysed through sonication, the suspension was centrifuged at 40,000 $g$ (JL-25.50 rotor, Beckman) for 30 min. The protein was purified by using the Ni-NTA column through the step-wise elution protocol with buffers containing 25-mM Tris-HCl (pH 7.5), 700-mM NaCl and 20, 50, or 300-mM imidazole. The fraction eluted in the buffer with 300-mM imidazole was pooled and concentrated to 15 mg mL⁻¹ in 30-kDa MWCO concentrator. The protein was further purified through gel filtration on a Superdex 200 Increase 10/300 GL column (GE Healthcare) in a buffer with 25-mM Tris-HCl (pH 7.5) and 700-mM NaCl. The major peak fractions were collected and concentrated to 15 mg mL⁻¹ for crystallization. The Se-Met protein was purified through the same procedure as the one used for native protein purification except that 2-mM DTT and 0.2-mM EDTA are added to the buffers. The detailed information about cell strains, plasmids, and primers used for cloning and protein expression is included in Supplementary Table 2.

**Crystallization of *Rt*MprF-SD and structure determination.** The protein crystals of the *Rt*MprF-SD were obtained through the hanging-drop vapor diffusion method at 16 °C with a well solution containing 0.1-M NaAc (pH 6.0) and 16% PEG3350. The Se-Met derivative crystals were grown with the well solution containing 0.1-M NaAc (pH 6.0) and 10% PEG3350. Both the native and the derivative crystals were cryo-protected in a solution with 0.1-M NaAc (pH 6.0), 10% PEG3350, and 20% glycerol before being flash-frozen in liquid nitrogen. The native and Se-Met derivative datasets were collected at 0.96000 and 0.97909-Å wavelength, respectively, on BL1A and NW12A beamlines in the Photon Factory (Tsukuba, Japan) using the UGUI v2 software, and were processed with the HKL2000 program[46]. The phases were solved through the single-wavelength anomalous method using Phenix AutoSol program[47]. Model building and structure refinement were accomplished by using Coot[48] and Phenix Refine. Structure figures were prepared with PyMOL (The PyMOL Molecular Graphics System, Version 2.0, Schrodinger, LLC).

**Reconstitution of *Rt*MprF in nanodiscs.** The purified *Rt*MprF protein was incorporated into lipid nanodiscs with a molar ratio of *Rt*MprF protein:membrane-scaffold-protein 1E3D1 (MSP1E3D1):POPG at 1:2:100. The mixture was incubated

at 4 °C for 1 h on a sample rotator. Reconstitution was initiated by removing detergent through addition of Bio-beads (Bio-Rad) to the sample and incubation at 4 °C overnight with constant rotation. In the next day, the old Bio-beads were replaced by fresh Bio-beads and the sample was further incubated for 2 h. Subsequently, Bio-beads were removed from the sample and the nanodisc reconstitution mixture was incubated with 0.25-mL Ni-NTA resin for 1 h at 4 °C to enrich nanodiscs with the target protein and remove the empty ones. The resin was washed with five column volumes of wash buffer (20-mM Tris-HCl pH 7.5, 300-mM NaCl, and 20-mM imidazole) followed by four column volumes of elution buffer (20-mM Tris-HCl pH 7.5, 300-mM NaCl, and 300-mM imidazole). The eluted *Rt*MprF protein in nanodiscs was further purified by loading the sample onto a Superdex 200 increase 10/300 GL size-exclusion column (GE Healthcare Life Sciences) and eluting it in the gel-filtration buffer with 20-mM Tris-HCl (pH 7.5) and 300-mM NaCl.

**Cryo-EM sample preparation and data acquisition.** The purified *Rt*MprF protein in nanodics was concentrated to 6 mg mL⁻¹ for *Rt*MprF(DDM)-nanodiscs or 13 mg ml⁻¹ for *Rt*MprF(GDN)-nanodiscs using a 100-kDa MWCO Amicon concentrator (Millipore). The Quantifoil 1.2/1.3-μm holey carbon grids (300 mesh, copper) were glow discharged for 60 s firstly and then 3 μl of concentrated nanodisc sample was applied onto the grid, blotted for 8.0 s with a force level of 2, drained for 0.5 s for *Rt*MprF(DDM)-nanodiscs, and blotted for 4.0 s with a force level of 4 for *Rt*MprF(GDN)-nanodiscs and plunge-frozen in liquid ethane cooled by liquid nitrogen, using Vitrobot Mark IV (Thermo Fisher) operated at 18 °C with 80% humidity for *Rt*MprF(DDM)-nanodiscs and 4 °C with 100% humidity for *Rt*MprF (GDN)-nanodiscs.

The grids containing *Rt*MprF nanodisc samples were imaged with a 200-kV Talos Arctica microscope equipped with a Gatan K2 Summit direct detector camera using the SerialEM acquisition software (ver. 3.6.2). An energy filter with slit width of 20 eV was used during data collection at a nominal magnification of ×130,000, resulting in a super-resolution pixel size of 0.5 Å (physical pixel size of 1.0 Å). Movies (32 frames per movie file) were captured with a defocus value at a range of −1.5 to −2.0 μm in the super-resolution mode using a dose rate of ~9.6 e⁻ pixel⁻¹ s⁻¹ over 5.2 s yielding a cumulative dose of ~50 e⁻ Å⁻².

**Image processing.** For *Rt*MprF(DDM)-nanodiscs, a total of 2921 cryo-EM movies were aligned with dose-weighting using MotionCor2 program[49] with 5 by 5 patches and a B-factor of 250. Micrograph contrast transfer function (CTF) estimations were performed by using CTFFIND 4.1.15 program[50]. Particle picking, 2D classification, and ab initio 3D reference generation were performed in cryoSPARC v2.9 program[51]. After manual inspection of the micrographs, 2549 were selected and ~100 particles were picked manually from the micrograph and sorted into 2D classes. The best classes were selected and used as references for subsequent autopicking procedure. After the process, 887,196 particles were auto-picked and extracted using a box size of 200 pixels. 2D classification was performed to remove ice spots, contaminants, and aggregates, yielding 529,371 particles. The particles were exported from cryoSPARC v2.9 using the UCSF pyem v0.5 script (https://doi.org/10.5281/zenodo.3576630) and re-extracted in RELION-3 program[52] from the original micrographs for 3D classification. Consequently, 247,321 particles were selected for further refinement. Per-particle CTF refinement, with estimation of the beam tilt and Bayesian polishing, was performed in RELION-3. Particles with resolution lower than 4-Å resolution were discarded, and the refinement with C2 symmetry imposed resulted in a 3.7-Å cryo-EM density map from a major class of 160,417 particles. For the minor classes of asymmetric shapes, four classes with tilted synthase domains are subject to a second round of 3D classification and the three classes with particle numbers over 10,000 are chosen for auto-refine with C1 symmetry individually. To improve the local map quality around LysPG in Cavity C, a mask covering Subdomain 1 was generated by Chimera and applied for local refinement in Relion 3. The local refinement procedure with the mask and solvent-flattened Fourier shell correlations yielded a reconstruction for Subdomain 1 at 3.4 Å.

For *Rt*MprF(GDN)-nanodiscs, a total of 2579 cryo-EM movies were aligned with dose-weighting using MotionCor2 program with 5 by 5 patches and a B-factor of 250. Micrograph CTF estimations were performed by using CTFFIND 4.1.15 program. Particle picking, 2D classification, and ab initio 3D reference generation were performed in cryoSPARC v2.9 program. After manual inspection of the micrographs, 2317 were selected and ~200 particles were picked manually from the micrograph and sorted into 2D classes. The best classes were selected and used as references for subsequent autopicking procedure. After the process, 1232,621 particles were auto-picked and extracted using a box size of 200 pixels. 2D classification was performed to remove ice spots, contaminants, and aggregates, yielding 343,117 particles. The particles were exported from cryoSPARC v2.9 using the UCSF pyem v0.5 script and re-extracted in RELION-3 program from the original micrographs for 3D classification. Consequently, 276,824 particles were selected, removed duplicates, and re-extracted with a box size at 320 instead of 200 pixels for further refinement. Per-particle CTF refinement, with estimation of the beam tilt and Bayesian polishing, was performed in RELION-3. A tight mask for TM domain was generated by Chimera and RELION-3, followed by 3D classification by skipping alignment. Finally, 144,479 particles were selected and the refinement with C2 symmetry imposed resulted in a 2.96-Å cryo-EM density map.

**Model building and refinement**. The structural model of the flippase domain of *Rt*MprF was built manually in Coot program[48], guided mainly by the cryo-EM map. The secondary structure prediction from PSIPRED Server[53] (ver. 4.0) and the transmembrane helix prediction from TMHMM Server[54] (ver. 2.0) were used as references during model building. While most of the transmembrane helices are identified in the map and the models are registered with amino acid sequences, the density for TM14 is too weak for MprF(DDM)-nanodiscs and it is tentatively interpreted with a poly-alanine α-helix model. For the synthase domain, the crystal structure was docked manually into the corresponding region of the cryo-EM map of the full-length *Rt*MprF, subject to rigid body refinement and local adjustment in Coot, and then merged with the flippase domain. The structural model of *Rt*MprF was refined against the cryo-EM map by using phenix.real_space_refine program followed by manual adjustment in Coot. The program refinement and manual adjustment were carried out iteratively till the model-to-map fitting is optimal and the model geometric parameters are within reasonable range (Supplementary Table 1). The final model covers 793 or 820 out of 869 amino acid residues of the full-length *Rt*MprF protein for MprF(DDM)-nanodiscs or MprF(GDN)-nanodiscs, respectively, while several regions in the loops or near the amino- and carboxyl-termini (1–22, 326–333, 375–384, 492–510, 531–538, and 861–869 region) for MprF(DDM)-nanodiscs or (1–23, 326–333, 531–539, 861–869 region) for MprF(GDN)-nanodiscs are unobserved in the map due to high flexibility.

**Crosslinking of *Rt*MprF**. The oligomeric state of the *Rt*MprF protein was analyzed through chemical crosslinking experiment by using the membrane preparation from the *E. coli* cells expressing the full-length *Rt*MprF. The cells were resuspended in a buffer consisting of 20-mM HEPES (pH 7.5) and 300-mM NaCl (buffer A). After the cells were lysed by passing through a high-pressure homogenizer (ATS Engineering), the cell debris was removed through low-speed centrifugation at 11,000 *g* for 15 min and the membrane fraction was further collected through ultracentrifugation at 100,000 *g* for 30 min at 4 °C. The membrane pellets were resuspended in buffer A and sonicated with 1 s on, 5 s off for 2 min to homogenize the sample. The membrane suspension was aliquoted and then treated with dis-uccinimidyl suberate (DSS) at 0–5-mM final concentration for 1 h at 30 °C with constant mixing on a shaker. The reactions were quenched by adding 100-mM Tris-HCl (pH 7.5). The crosslinked samples were solubilized by adding 1% β-DDM for 1 h in the shaker. Subsequently, the samples were centrifuged at 18,000 *g* for 10 min and the supernatant was mixed with 5 × SDS-PAGE loading buffer, and then loaded on the SDS-PAGE gel for electrophoresis. The protein bands on the gel were transferred to polyvinylidene difluoride membrane and then detected through western blot by using the Anti-His Mouse monoclonal antibody (1:80,000 dilution) and Goat anti-Mouse IgG (H + L)–HRP (1:20,000 dilution). After being developed with the western lightning Ultra ECL horseradish peroxidase substrate (Perkin–Elmer), the blots were imaged on a chemiluminescence CCD system (ChemiScope 3500 mini imager, Clinx Science Instruments).

**TLC and mass spectrometry**. The lipids from *E. coli* membrane expressing recombinant *Rt*MprF/*RtPa*MprF or from the purified *Rt*MprF protein samples were extracted according to Bligh and Dyer procedure[55]. In detail, 12-mL chloroform:methanol (1:2, v:v) mixture was added to 3.2-mL sample and mixed well through vortex. Subsequently, 4-mL chloroform was added to the sample and vortexed again to mix. Finally, 4-mL water was added to the sample and vortexed well. The mixture was centrifuged at 100 *g* for 5 min to get a two-phase system with aqueous phase at the top and organic phase at the bottom. The bottom phase was washed twice with an aqueous upper phase solution (freshly made by mixing chloroform, methanol, and water at 2:2:1.8 (v:v:v) ratio and centrifuging the mixture). Finally, the bottom phase was recovered, dried under vacuum, and dissolved in 100-μL chloroform. The lipid samples were separated on the HPTLC silica gel 60 F254 plates (Merck) in a mobile phase of chloroform:methanol:water mixture (65:25:4). Lipid spots were visualized through staining with iodine or ninhydrin. For separation of AlaPG and PE, a mobile phase of chloroform:methanol:acetic acid:water (80:12:15:4) mixture was used.

The liquid chromatography–mass spectrometry (LC-MS)/MS analysis was performed by using a Thermo Scientific Dionex Ultimate 3000 LC system coupled to a TripleTOF 5600 quadrupole time-of-flight tandem mass spectrometer. An ACQUITY UPLC C18 reversed-phase column (1.7 μm, 2.1 × 100 mm, Waters) was used in LC. Mobile phase A consisted of methanol/acetonitrile/aqueous 15-mM ammonium acetate (1:1:1, vol/vol/vol), and mobile phase B consisted of 80% 2-propanol and 20% methanol containing 5-mM ammonium acetate. The LC process was operated at a flow rate of 250 μL min[−1] with a linear gradient as follows: 10% B was held constantly for 1 min and then increased linearly to 60% B over 5 min, further to 100% B over 12 min and finally held at 100% B for 2 min. The conditions for MS were set with the following parameters: electrospray voltages, +5500 V (positive ion mode) and −4400 V (negative ion mode); declustering potential, 100 V; GS1 and GS2, 60 psi. The collision-induced dissociation tandem mass spectra were obtained with collision energy of +35 V in the positive ion mode or −35 V in the negative ion mode. Nitrogen was used as the collision gas.

For quantification of lipid:protein molar ratio, the lipids were extracted from 47.5 nmol of purified *Rt*MprF(DDM) protein according to Bligh and Dyer procedure and were dissolved in 100-μL chloroform. Two microliter of the lipid

solution were applied on the HPTLC silica gel 60 F254 plates (Merck) and separated in the solvent of chloroform:methanol:water mixture (65:25:4). As the standard samples, 0.25, 0.5, 1.0, 2.0, 4.0-nmol LysPG (Avanti) were also loaded on the same plate. For quantification of LysPG:protein stoichiometry in *Rt*MprF (GDN), 17.8-nmol protein was used and extracted in 40-μl chloroform. The standard samples of 1.0, 2.0, 4.0, and 8.0-nmol LysPG were loaded on the plate as references. Lipid spots were visualized through the iodine staining procedure. It is noteworthy that the data obtained through iodine staining generates a linear standard curve better than those stained with ninhydrin. For quantifying the relative amount of LysPG/AlaPG co-purified with *Rt*MprF/*RtPa*MprF mutants, lipids extracted from same amount of purified *Rt*MprF/*RtPa*MprF mutant protein were separated on HPTLC silica gel 60 F254 plates (Merck) and stained by ninhydrin. The protein concentration was measured through the bicinchoninic acid method (TransGen Biotech, Beijing). The mobile phase chloroform:methanol:water mixture (65:25:4) and chloroform:methanol:acetic acid:water (80:12:15:4) were utilized to separate lipids extracted from *Rt*MprF and *RtPa*MprF mutant protein, respectively. The image of the iodine- or ninhydrin-stained TLC plate was processed by Image J and analyzed by the GraphPad program. For the data presented in Figs. 3e, f, 4c and Supplementary Fig. 9b, f, three aliquots of the sample of the same type are loaded on the TLC plates and the measurements of the three parallel spots are used for statistical analysis.

**Relative quantification of total LysPG and fluorescamine-labeled LysPG**. The cells were cultured in Terrific Broth media containing 50-μg mL[−1] ampicillin, and protein expression was induced with 0.5-mM IPTG for 2 h at 37 °C after $OD_{600}$ reached 1.0. After the cells were harvested through centrifugation, the pellets were washed and suspended in a solution containing 25-mM HEPES-Na (pH 8.5) and 300-mM NaCl (Buffer B), and then adjusted to a concentration of $4 \times 10^9$ cell mL[−1].

For total lipid extraction, 5-mL cell suspension was centrifuged and resuspended in 1.2-mL Buffer B. Subsequently, 4.5-mL chloroform:methanol (1:2, v:v) mixture was added to the suspension and vortexed for 30 s. Afterwards, 1.5-mL chloroform and 1.5-mL HEPES buffer were added to the sample sequentially and vortexed for 10 s after each step. After the mixture was centrifuged at 100 *g* for 5 min, the bottom phase was recovered, dried under vacuum, and dissolved in 150-μl chloroform. To separate LysPG from other lipids, 4.5-μL total lipid samples were loaded on the HPTLC plate and the plate was developed in a solvent of chloroform:methanol:water (65:25:4) mixture, dried, and then stained with ninhydrin.

For detection of LysPG in the outer leaflet of the membrane, 50-μL fluorescamine (50-mM stock solution in DMSO) was added to 5-mL cell suspension and incubated at room temperature for 20 min. Afterwards, 500-μL Tris-HCl (pH = 8.0, 1 M) was added to the mixture and incubated for 5 min to stop the reaction. Finally, the cells were collected through centrifugation, washed once in 5-mL Buffer B and resuspended in 1.2-mL Buffer B. The following procedures of lipid extraction and TLC experiments were the same with the above protocols used for the total lipid extraction sample, except that the lipid spots were visualized under UV light after they were separated on HPTLC plates. Image analysis was accomplished by using Image J and GraphPad program. For the data presented in Fig. 4d–k, the aliquots of three repeats of distinct samples are loaded on the TLC plates and the measurements of the three parallel spots are used for statistical analysis. For Supplementary Fig. 9e, the data presented are mean values of three independent repeats of TLC experiments with distinct samples.

**Computational modeling analysis**. The model of *Sa*MprF is constructed through the Modeller 9.23 program[38] by using the cryo-EM structure of *Rt*MprF as the template and the amino acid sequence alignment data of the two homologs as the other input. Virtual docking of daptomycin molecule on *Sa*MprF was carried out through the Autodock Vina program (v1.1.2)[56] by providing the homologous model of *Sa*MprF and the structure of daptomycin downloaded from the Protein Data Bank [PDB code: 1T5M (https://www.rcsb.org/structure/1T5M)]. A cubic box with 60 × 60 × 60 grid points (in the *x*, *y*, and *z* dimensions) and 0.375-Å spacing was applied to define the search region on the *Sa*MprF model during the Autodock analysis.

**Reporting summary**. Further information on research design is available in the Nature Research Reporting Summary linked to this article.

## Data availability
Data supporting the findings of this manuscript are available from the corresponding author upon reasonable request. A reporting summary for this Article is available as a Supplementary Information file. The coordinates and the cryo-EM maps for *Rt*MprF (DDM)-nanodiscs and *Rt*MprF(GDN)-nanodiscs have been deposited in the PDB [6LVF and 7DUW] and EMDB [EMD-0992 and EMD-30869]. The crystal structure and the structure factors for *Rt*MprF-SD have been deposited in the PDB [6LV0]. The other PDB files used for analysis and comparison in this work were downloaded from the RCSB PDB website (http://www.rcsb.org/), including the structures of daptomycin [accession code: 1T5M], FemX-aminoacyl-tRNA complex [accession code: 4II9], *Bl*MprF-LYN complex structure [accession code: 4V36], TMEM16F [accession code: 6QP6], MurJ

[accession code: 6NC9], MsbA [accession code: 5TV4], and Drs2p-Cdc50p [accession code: 6ROJ]. Source data are provided with this paper.

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

## Acknowledgements

We thank J. J. Wu for his initial efforts in cloning and purifying various MprF homologs. The cryo-EM data were collected at the Center for Biological Imaging (CBI), Core Facilities for Protein Science at the Institute of Biophysics, Chinese Academy of Sciences, and the beamtime on Talos Arctica was sponsored by the National Laboratory of Biomacromolecules and CBI. We also thank X. J. Huang, B. L. Zhu, L. H. Chen, and F. Sun at CBI for their assistance in cryo-EM data collection; Z. S. Xie for her technical assistance on the mass spectrometry analysis of lipid samples; the staffs at the Photon Factory (BL1A and NW12A), Shanghai Synchrotron Radiation Facility (BL17U and BL18U1), and the Protein Science Research Facility of IBP (Y. Han) for their assistance on crystal screening and X-ray data collection; X. B. Liang for her support in sample preparation, data collection, and storage. The project is funded by the National Natural Science Foundation of China (31670749 and 31925024), the Basic Frontier Science Research Program of CAS (ZDBS-LY-SM003-02), and the Strategic Priority Research Program of CAS (XDB37020101 and XDB08020302).

## Author contributions

D.S. purified the *Rt*MprF protein, reconstituted the nanodisc sample, prepared cryo-EM grids, collected cryo-EM data, processed the data, built the model and refined the structure, and performed quantitative analysis on the proportion of LysPG in total phospholipids extracted from *E. coli* cells through the TLC method. H.J. optimized the protocol for *Rt*MprF expression and purification, solved the crystal structure of *Rt*MprF synthase domain, performed site-directed mutagenesis, and carried out lipid extraction and TLC experiments. D.S., H.J., and Z.L. analyzed the structure and wrote the manuscript. Z.L. conceived and supervised the project.

## Competing interests

The authors declare no competing interests.
