## [Peer Review File · Nature Communications]

REVIEWER COMMENTS

Reviewer #2 (Remarks to the Author):

The authors made significant progress with the manuscript, which is now in a suitable form. All my concerns were appropriately addressed. I regard the study as an outstanding advance with the first structure of a bacterial phospholipid flippase. Moreover, the study may help to target MprF-like proteins with new drugs and understand how MprF may contribute to daptomycin resistance. I have only some minor comments:

- I was puzzled by the additional figures without legends beyond page 72. Are these old versions from the initial submission? I hope I did not overlook something important here.
- It would be helpful to restructure the text with more paragraphs.
- It could also help to indicate the identity of phospholipids in Fig. 4D, F, H, J and Extended Figures 9, as in Fig. 4A.

Reviewer #3 (Remarks to the Author):

The Multiple Peptide Resistance Factors (MprFs) are a family of bacterial proteins that play important roles in resistance to antibiotics and antimicrobial peptides. They contain two domains: cytosolic aminoacyl phosphatidylglycerol (aaPG) synthase domain and membrane-embedded aaPG flippase domain. While isolated aaPG synthase domain structure has been determined, the structure and mechanism of the entire MprF is not well understood. This manuscript presents two cryo-EM structures of MprF in nanodiscs, initially purified in DDM or GDN detergent. The structural findings were also characterized using biochemical assays. Overall, the cryo-EM studies were nicely performed, and most conclusions are supported by the structural and biochemical data.

This reviewer has a few comments on the interpretation of cryo-EM maps, particularly the newly added MprF(GDN)-nanodisc map and the potential lipid densities. The lipids in C site and P site and at dimer interface are crucial for this manuscript, and thus should be carefully interpreted. The following points should be addressed:

- (1) C site. The lipid density in C site has three arms, with the one mostly outside being the weakest. This three-arm density per se does not have sufficient resolution to tell which arm is Lys head group. The density assignment of Lys headgroup is critically supported by the local protein side-chain environment, mutagenesis, and biochemical studies. These points should be clearly presented where C site and its lipid substrate are presented. In the MprF(GDN)-nanodisc map, right below the C-site LysPG, there is a diacyl lipid density. Is this worth mentioning as a potential site for PG capture or LysPG release from the synthase domain?
- (2) P site. There is only noise in the P site of MprF(DDM)-nanodisc map. In the P site of MprF(GDN)-nanodisc map, there are three well-resolved acyl chains, but no clear density to conclude the presence or absence of Lys head group. The authors should make these clear and revise the interpretation.

(Even LysPG is not conclusively visualized in the cryo-EM map, the well-ordered acyl chains strongly support the potential binding of PG or LysPG.) The authors should also explain how the three acyl chains were modeled.

(3) Homodimer interface. In MprF(GDN)-nanodisc map, the upper modeled LysPG is occupying a clearly flat density, which cannot be an acyl chain but more likely the flat ring structure in GDN. This flat density is next to a density with two acyl chains (of PG). Note that, this model is fine in the MprF(DDM) map, because the corresponding density has a tube shape for an acyl chain. Presumably the authors did not pay attention to the difference of the densities between these two maps, and directly used the same lipid model from MprF(DDM)-nanodisc.

This reviewer also checked authors' responses to the comments from previous Reviewer #1:

1. All addressed.
2. The newly added cryo-EM structure of MprF(GDN) significantly strengthened the manuscript. Please make it clear in the text that, only in this new map, P site contains well-resolved acyl chains, of PG or LysPG.
3. All addressed.
4. See (2) above. There is no clear density to support LysPG head group in the P site. The zoomed and masked-out density as shown in this figure is misleading. Instead, a map without modification should be presented with display (sigma) level indicated.

Point-to-point Responses

Responses to Reviewer #2's comments

The authors made significant progress with the manuscript, which is now in a suitable form. all my concerns were appropriately addressed. I regard the study as an outstanding advance with the first structure of a bacterial phospholipid flippase. Moreover, the study may help to target MprF-like proteins with new drugs and understand how MprF may contribute to daptomycin resistance.

Response: Thank you very much for your positive and supporting comments. We are grateful for your insightful advices which helped us greatly in improving our work.

- I was puzzled by the additional figures without legends beyond page 72. Are these old versions from the initial submission? I hope I did not overlook something important here.

Response: The additional figures without legends (page 73-88) are high-resolution versions of Figures 1-5 and Extended Figures 1-11. The contents of these figures are identical to those shown in page 42-71 with legends. (They are not the old versions from the initial submission.) Sorry for the duplication and confusion. These figures without legends were submitted along the manuscript and integrated with the manuscript automatically while they were converted into a single pdf files by the manuscript processing server.

- It would be helpful to restructure the text with more paragraphs.

Response: Yes, thanks a lot for the nice suggestion. In the revised manuscript, we have reorganized the texts by dividing the long paragraphs of lines 183-248 into three paragraphs (p9-11), lines 249-279 into two paragraphs (p11-12), lines 281-306 (p12-13) into two paragraphs, lines 309-341 (p14-15) into three paragraphs, lines 376-407 (p16-18) into two paragraphs and lines 533-578 (p23-24) into two paragraphs.

- It could also help to indicate the identity of phospholipids in Fig. 4D, F, H, J and Extended Figures 9, as in Fig. 4A.

Response: Thank you for the suggestion. We have labeled the identity of phospholipids in the updated version of Fig. 4d, f, h, j and Extended Fig. 9b, d &f. (Note: the Extended Data Figures have been changed to Supplementary Figures in the revised manuscript according to the latest format style of *Nat. Commun.* journal).

Responses to Reviewer #3's comments:

The Multiple Peptide Resistance Factors (MprFs) are a family of bacterial proteins that play important roles in resistance to antibiotics and antimicrobial peptides. They contain two domains: cytosolic aminoacyl phosphatidylglycerol (aaPG) synthase domain and membrane-embedded aaPG flippase domain. While isolated aaPG synthase domain structure has been determined, the structure and mechanism of the entire MprF is not well understood. This manuscript presents two cryo-EM structures of MprF in nanodiscs, initially purified in DDM or GDN detergent. The structural findings were also characterized using biochemical assays. Overall, the cryo-EM studies were nicely performed, and most conclusions are supported by the structural and biochemical data.

Response: Thank you very much for your positive comments and very helpful advices.

This reviewer has a few comments on the interpretation of cryo-EM maps, particularly the newly added MprF(GDN)-nanodisc map and the potential lipid densities. The lipids in C site and P site and at dimer interface are crucial for this manuscript, and thus should be carefully interpreted. The following points should be addressed: (1) C site. The lipid density in C site has three arms, with the one mostly outside being the weakest. This three-arm density per se does not have sufficient resolution to tell which arm is Lys head group. The density assignment of Lys headgroup is critically supported by the local protein side-chain environment, mutagenesis, and biochemical studies. These points should be clearly presented where C site and its lipid substrate are presented.

Response: We thank the reviewer for the excellent point about modeling of the lipid density in C site. In the revised manuscript, we have added a few sentences in lines 187-199, p9 to discuss the three-arm density features of LysPG1 and the evidences for assignment of lysyl head group and two fatty acyl groups of LysPG1. “The density of LysPG1 exhibits three arms with similar shape and length (Supplementary Fig. 3b). The first arm is buried inside Cavity C and surrounded by polar amino acid residues,

such as Asn117, Asp234, Ser238 and Arg304. The second arm is sandwiched between TM7a and TM4, and surrounded by hydrophobic residues. The density of the third arm is the weakest among the three and it is located at the outmost region exposed to the hydrophobic area of lipid bilayer. Although the local resolution of the three individual arms may appear insufficient for distinguishing the phospho-[3-lysyl(1-glycerol)] head group and two fatty acyl groups, interpretation of the lipid molecule is assisted by considering the compatibility of the individual groups with their local environments. As a result, the first arm is assigned as the phospho-[3-lysyl(1-glycerol)] head group and the other two arms most likely belong to the fatty acyl chains of LysPG molecule. The model is further verified through mutagenesis and biochemical analysis (described below).”.

In the MprF(GDN)-nanodisc map, right below the C-site LysPG, there is a diacyl lipid density. Is this worth mentioning as a potential site for PG capture or LysPG release from the synthase domain?

Response: As the reviewer mentioned, there is indeed a lipid-like density nearby the C-site LysPG1 (Response Fig. 1).

Response Figure 1. A peripheral lipid-like density nearby LysPG1 and at the cytoplasmic entrance of Cavity C. The MprF(GDN)-nanodisc map is contoured at $1.2 \times \sigma$ level.

The density is sandwiched by LysPG1 and the β -hairpin loop between TM5 and TM6. As the head group density is too weak to allow unambiguous identification of the lipid, we could not interpret the density with certainty. Therefore, no reliable model can be

built for this density feature. As this lipid is located at the entrance of Cavity C, it may serve as a potential site for LysPG loading after it is synthesized and released from the synthase domain. Alternatively, in case that a PG molecule is captured at this site, it may serve as the substrate for the synthase domain if it could diffuse laterally to the active site of the synthase domain at a horizontal position close to the membrane surface. The third possibility is that it may belong to the bulk lipid (such as phosphatidylethanolamine/PE or elses) from the membrane, serving to stabilize LysPG1 in Cavity C and prevent unloading of LysPG1 on the cytoplasmic side by blocking the cytoplasmic portal of Cavity C. A short paragraph has been added in the revised manuscript to discuss about the topic (lines 486-496, p21) and the Response Figure 1 is included in the manuscript as Supplementary Fig. 12.

(2) P site. There is only noise in the P site of MprF(DDM)-nanodisc map. In the P site of MprF(GDN)-nanodisc map, there are three well-resolved acyl chains, but no clear density to conclude the presence or absence of Lys head group. The authors should make these clear and revise the interpretation. (Even LysPG is not conclusively visualized in the cryo-EM map, the well-ordered acyl chains strongly support the potential binding of PG or LysPG.) The authors should also explain how the three acyl chains were modeled.

Response: Thank you for the suggestion. For the lipid molecule in Cavity P (or P site) of MprF(GDN)-nanodisc, there are indeed three well-resolved fatty acyl chains inside the cavity as the reviewer mentioned (Response Figure 2a). Among them, the two long ones join each other at the head group region near periplasmic surface and belong to a phospholipid molecule tentatively assigned as LysPG2 in the model. The third acyl chain is much shorter than the other two, likely belonging to a detergent molecule (DDM, from the early solubilization step). While the density for the two fatty acyl chains of the lipid molecule are fairly strong (clearly visible at $1.2-2.0 \times \sigma$ level), the head group density is relatively weak (Response Figure 2b). When the contour level is lowered to $0.8 \times \sigma$, the density corresponding to the lysyl group becomes visible and appears to be connected to the glycerol group. Therefore, the lipid density feature in Cavity P is interpreted as a LysPG molecule with highly flexible lysyl group. Alternatively, a PG molecule may also occupy the site.

Response Figure 2. The cryo-EM densities for the lipid and detergent molecules in Cavity P of MprF(GDN)-nanodisc. a, The map is contoured at $1.2 \times \sigma$ level and shows three strong fatty acyl chains in Cavity P. **b,** The density feature potentially belonging to the lysyl group becomes visible when the map contour level is lowered at $0.8 \times \sigma$.

Response Figure 3. Quantification of the content of LysPG co-purified with *RtMprF(GDN)* protein through TLC method. The TLC plate was stained by iodine and the image shown in a was processed by ImageJ. The data extracted from the spots of LysPG standard samples were used for linear regression of the standard curve ($Y = 1202 \times X + 2197$) as shown in b. By referring to the standard curve, the amount of LysPG extracted from 0.89 nmol *RtMprF* protein purified in GDN is estimated to be 2.340 (corresponding to ~ 2.6 LysPG per *RtMprF* monomer) as indicated by the red label. The error bars in b denote the standard errors of the mean values ($n=3$).

Meanwhile, we have also measured the content of LysPG co-purified with *RtMprF*(GDN) protein through the thin-layer chromatography (TLC) method (Response Figure 3). The stoichiometry of LysPG in the *RtMprF*(GDN) protein is estimated to be ~2.6 LysPG molecules per *RtMprF* monomer, much higher than the stoichiometry of LysPG in the *RtMprF*(DDM) protein (~1.2). Therefore, it is highly possible that a second LysPG binding site does exist in *RtMprF* and the interpretation of the lipid density in Cavity P as a LysPG is consistent with the LysPG:*RtMprF* stoichiometry data.

A few sentences are added in the revised manuscript (lines 232-243, p11; lines 261-266, p12) to explain the details about interpretation and modeling of the lipid and detergent densities in the Cavity P of *RtMprF*(GDN)-nanodisc. The Response Figures 2 and 3 are included as supplementary Fig. 8 and supplementary Fig. 9f&g in the revised manuscript.

(3) Homodimer interface. In *MprF*(GDN)-nanodisc map, the upper modeled LysPG is occupying a clearly flat density, which cannot be an acyl chain but more likely the flat ring structure in GDN. This flat density is next to a density with two acyl chains (of PG). Note that, this model is fine in the *MprF*(DDM) map, because the corresponding density has a tube shape for an acyl chain. Presumably the authors did not pay attention to the difference of the densities between these two maps, and directly used the same lipid model from *MprF*(DDM)-nanodisc.

Response: Thank you very much for the insightful advice. We have tried to fit the model of a GDN molecule into the density you mentioned (Response Fig. 4). Indeed, it appears to match well with the density, much better than the previous model of a phospholipid molecule. Therefore, a GDN molecule is identified at the dimerization interface and the two fatty acyl chain densities nearby GDN are assigned to a PG molecule (PG3). The Supplementary Fig. 3c has been updated accordingly, and the revised pdb file and a new validation report of *MprF*(GDN)-nanodisc (*RtMprF_GDN_real_space_refined.pdb* and *D_1300019269_val-report-full_P1.pdf*) have been uploaded along with the revised manuscript. A sentence has been added in the revised manuscript to describe the observation (line 118-121, p6). “In *RtMprF*(GDN)-nanodisc, the hydrophobic group of a GDN molecule occupies the

binding site of the 2-acyl chain of PG2 molecule observed in *RtMprF*(DDM)-nanodisc, while the other PG molecule (PG3) at the peripheral region of the dimerization interface is located nearby GDN (Supplementary Fig. 3c).”.

Response Figure 4. Fitting of the GDN and PG models in the densities at the dimerization interface of *RtMprF*(GDN)-nanodisc. The cryo-EM map is contoured at $1.0\times\sigma$ level.

This reviewer also checked authors’ responses to the comments from previous Reviewer #1: 1. All addressed. 2. The newly added cryo-EM structure of *MprF*(GDN) significantly strengthened the manuscript. Please make it clear in the text that, only in this new map, P site contains well-resolved acyl chains, of PG or LysPG.

Response: Thanks for your advice. We have added a sentence in line 246-248 (p11) to clarify the point. “Such a well-resolved lipid feature in Cavity P is only present in *RtMprF*(GDN)-nanodisc but not in *RtMprF*(DDM)-nanodisc.”

3. All addressed. 4. See (2) above. There is no clear density to support LysPG head group in the P site. The zoomed and masked-out density as shown in this figure is misleading. Instead, a map without modification should be presented

with display (sigma) level indicated.

Response: As discussed above, the density for the head group of LysPG is relatively weak compared to those of acyl chains, but is visible at low contour level ($0.8 \times \sigma$, Response Figure 2). In the revised manuscript, we have included the map figures shown in Response Figure 2 (without modification) in the Supplementary Figure 8 as supporting evidence and the corresponding sigma (contour) levels are described in the legend.

REVIEWERS' COMMENTS

Reviewer #3 (Remarks to the Author):

This revised version of manuscript has adequately addressed all the points I raised, particularly on interpretation of cryo-EM maps. It is now suitable for publication. This is an excellent work studying the detailed mechanism of a phospholipid modification and transport protein machine.